# The histone demethylase Phf2 acts as a molecular checkpoint to prevent NAFLD progression during obesity

Julien Bricambert[1,2,3], Marie-Clotilde Alves-Guerra[1,2,3], Pauline Esteves[1,2,3], Carina Prip-Buus[1,2,3], Justine Bertrand-Michel[4], Hervé Guillou [5], Christopher J. Chang[6,7], Mark N. Vander Wal[6], François Canonne-Hergaux[8,9,10,16], Philippe Mathurin[11,12,16], Violeta Raverdy[13,14,15], François Pattou [13,14,15], Jean Girard[1,2,3], Catherine Postic[1,2,3] & Renaud Dentin [1,2,3]

Aberrant histone methylation profile is reported to correlate with the development and progression of NAFLD during obesity. However, the identification of specific epigenetic modifiers involved in this process remains poorly understood. Here, we identify the histone demethylase Plant Homeodomain Finger 2 (Phf2) as a new transcriptional co-activator of the transcription factor Carbohydrate Responsive Element Binding Protein (ChREBP). By specifically erasing H3K9me2 methyl-marks on the promoter of ChREBP-regulated genes, Phf2 facilitates incorporation of metabolic precursors into mono-unsaturated fatty acids, leading to hepatosteatosis development in the absence of inflammation and insulin resistance. Moreover, the Phf2-mediated activation of the transcription factor NF-E2-related factor 2 (Nrf2) further reroutes glucose fluxes toward the pentose phosphate pathway and glutathione biosynthesis, protecting the liver from oxidative stress and fibrogenesis in response to diet-induced obesity. Overall, our findings establish a downstream epigenetic checkpoint, whereby Phf2, through facilitating H3K9me2 demethylation at specific gene promoters, protects liver from the pathogenesis progression of NAFLD.

[1] INSERM, U1016, Institut Cochin, Paris, France. [2] CNRS, UMR8104 Paris, France. [3] Université Paris Descartes, Sorbonne Paris Cité, Paris, France. [4] Plateau de lipidomique, Bio-Medical Federative Research Institute of Toulouse, INSERM, Plateforme MetaToul, Toulouse, France. [5] INRA-ToxAlim, Toxicologie Intégrative et Métabolisme, Toulouse, France. [6] Department of Chemistry and Molecular and Cell Biology, University of California, Berkeley, CA 94720, USA. [7] Howard Hughes Medical Institute, University of California, Berkeley, CA 94720, USA. [8] INSERM U1043-CPTP, Toulouse F-31300, France. [9] CNRS, U5282, Toulouse F-31300, France. [10] Université de Toulouse, UPS, Centre de Physiopathologie de Toulouse Purpan (CPTP), Toulouse F-31300, France. [11] Department of Hepatology, Lille University Hospital, Lille, France. [12] Inserm, U 995 Lille, France. [13] INSERM, U859 Biotherapies for Diabetes, Lille, France. [14] European Genomic Institute for Diabetes, Lille University, Lille, France. [15] Department of Endocrine Surgery, Lille University Hospital, Lille, France. [16]Present address: Institut de Recherche en Santé Digestive (IRSD), Université de Toulouse, ENVT, INPT, INRA UMR1416, INSERM UMR1220, UPS, Toulouse, France. Correspondence and requests for materials should be addressed to R.D. (email: renaud.dentin@inserm.fr)

Non Alcoholic Fatty Liver Disease (NAFLD), one of the most prevalent metabolic disorders worldwide[1,2], is characterized by multiple distinct injurious mechanisms, which are unified under the concept of multiple parallel hits hypothesis[3]. All of these factors, including accumulation of lipotoxic intermediates, inflammation, oxidative stress, hepatocyte apoptosis, and fibrogenesis, act in a complex way to enhance the development and progression of the hepatic lesions through the NAFLD spectrum[3]. Although various genetic factors contribute to NAFLD development[4], it is now widely accepted that environmental factors, by directly altering the epigenome, are also key determinants[5,6]. Overall, excessive nutrient intake precipitates dynamic epigenetic modifications in some specific pattern of gene expression, causing an individual to become prone to diet-related metabolic disorders[7–11]. The fact that aberrant histone methylation profiles can be associated with metabolic syndrome reinforces the concept that histone lysine methylation state has a central role in this process[12,13]. Accordingly, the expression of specific histone lysine methyltransferases (KMT) and demethylases (KDMs) is altered during hepatosteatosis development[14]. Although these findings suggest that epigenomic control is crucial for NAFLD development, the exact nature of those epigenetic modifiers remains partially unknown[9].

In this study, we identify the histone demethylase plant homeodomain finger two (Phf2), which belongs to the KDM7 histone demethylase family, as a component of the ChREBP-assembled transcriptional complex, a transcription factor previously implicated in hepatic steatosis development[15,16]. While Phf2 has been demonstrated to play roles in glucose metabolism in hepatocytes and pro-inflammatory response in macrophages using in vitro experimental approaches, the physiological roles of Phf2 are still unclear, partially because Phf2 is ubiquitously expressed in various metabolic tissues and appears to work as a coactivator with multiple transcription factors. Supporting its potential metabolic function, it has been recently shown that Phf2 also controls adipogenesis and fat storage through the regulation of CEBPα and PPARγ transcriptional activities in adipose tissue[17,18]. Indeed, mice with targeted disruption of Phf2 or Arid5b (AT-rich interactive domain D), a specific Phf2 coactivator partner[19,20], display a reduction of white adipose tissue mass[21] as a result of reduced PPARγ activity[22]. Overall, it seems that Phf2 could work in multiple organs as a determinant regulator of glucose and lipid homeostasis. However, the regulation of Phf2 activity in hepatocytes and its contribution to the physiopathology of obesity and type 2 diabetes and more specifically to the development and/or progression of NAFLD are currently unknown.

Our study highlights, both in mice and human, a role for active Phf2-mediated H3K9me2 demethylation as a molecular checkpoint in the regulation of a subset of metabolic and anti-oxidative gene programs by interfering with the activity of the transcription factors ChREBP and NF-E2-related factor 2 (Nrf2). As a consequence, Phf2 activation protects the liver from inflammation, oxidative stress, insulin resistance, and fibrosis development during the pathogenesis progression of NAFLD during obesity. These findings link Phf2 to sugar sensing in hepatocytes and may allow for novel therapeutic approaches to treat related-metabolic liver disorders.

## Results

### Identification of Phf2 as a ChREBP interacting protein. During identification of transcriptional co-regulators for the transcription factor ChREBP in hepatocytes using mass spectrometry analysis, the jmjC domain containing histone demethylase Phf2, which is highly expressed in the liver, was purified (Supplementary Fig. 1a

and b). This association was confirmed by reciprocal co-immunoprecipitation studies using either ectopically expressed proteins in 293T cells or with endogenous proteins in the liver of fed C57Bl/6J mice (Supplementary Fig. 1c). In the search for Phf2 molecular function, FLAG-tagged Phf2, immunoprecipitated from 293T cells, demethylated H3K9me2 histone methyl mark on recombinant histones, core histones or mono-nucleosomes, but had no effect on other histone methylation marks tested (Fig. 1a, b and Supplementary Fig. 1d). This specific histone demethylase activity was further confirmed by immunostaining of H3K9me2 histone mark in Phf2 overexpressing hepatocytes (Fig. 1c). Accordingly, mutation of histidine 248, predicted to be part of the Fe(II) binding site of Phf2 jmjC domain[23], impaired this activity (Fig. 1b). ChIP-sequencing (ChIP-seq) demonstrated that Phf2 peaks (~79%) predominantly localized and covered the transcription start site (TSS) on its target gene promoters (Fig. 1d, e). Correlating with Phf2 tags distribution, the intensity of H3K9me2 methyl marks was predominantly decreased at the TSS of Phf2-bound promoters, further supporting its specific H3K9me2 demethylase activity (Fig. 1e, right panel). The majority of Phf2-bound promoters (90%) were also H3K4me3 and RNA-polII positive, suggesting that they were either "poised" for transcription or transcriptionally active (Fig. 1d, e). Accordingly, in vitro peptide pull down assay showed that Phf2 bound specifically to H3K4me3 histone tails (Fig. 1f). In contrast, a Phf2 mutant lacking its methyl lysine-binding PHD domain (Phf2ΔPHD) did not bind H3K4me3 histone tails, and more importantly did not retain its histone demethylase activity toward mono-nucleosomes (Fig. 1f). This reveals that the PHD finger-mediated targeting of Phf2 to H3K4me3-containing nucleosomes is required to demethylate H3K9me2 methyl marks. Bioinformatics analysis also revealed that the most statistically predicted Phf2 binding sites were E-box and ARE-response elements (Fig. 1g). Accordingly, gene ontology analysis of Phf2-bound promoters indicated that Phf2 was recruited to the promoter of genes involved in the regulation of metabolic processes, cell cycle and response to oxidative stress (Fig. 1h). Supporting this observation, microarray analysis, conducted on Phf2 overexpressing hepatocytes, demonstrated that Phf2 affected the expression of specific transcriptional networks under the control of several key metabolic transcription factors, such as ChREBP, PPARγ, Nrf2, or HIF1α (Fig. 1i).

### Phf2 activation promotes NAFLD development. To evaluate its function, Phf2 was overexpressed for 3 weeks in the liver of C57BL/6J mice. Phf2 overexpression led to hepatic steatosis development as shown by increased liver weight (Supplementary Table 1) and by oil red O staining of liver sections (Fig. 2a). Microarray analysis demonstrated that Phf2 overexpression led to the differential regulation of 921 genes (fold ± 1.5 and $p < 0.05$). Among them, 25% were involved in the control of metabolic processes (Supplementary Fig. 2a–d). In this context, metabolic tracing studies, using $^{14}C$-labeled glucose and oleate as substrates, demonstrated that Phf2 overexpression increased hepatic fatty acid (FA) uptake, esterification and de novo lipogenesis (DNL) rates (Supplementary Table 1). In agreement, expression of genes that promote FA uptake and activation, in addition to their intracellular trafficking and esterification was enhanced in Phf2 mice (Supplementary Fig. 3a, b). Expression of genes involved in DNL and lipid sequestration was also upregulated (Supplementary Fig. 3a, b). Finally, hepatic TG secretion rate was also decreased in Phf2 mice correlating with reduced expression of MTP and ApoB involved in VLDL assembly and secretion (Supplementary Fig. 3a–c). Overall, Phf2 overexpression by controlling metabolic rerouting and lipid sequestration favors

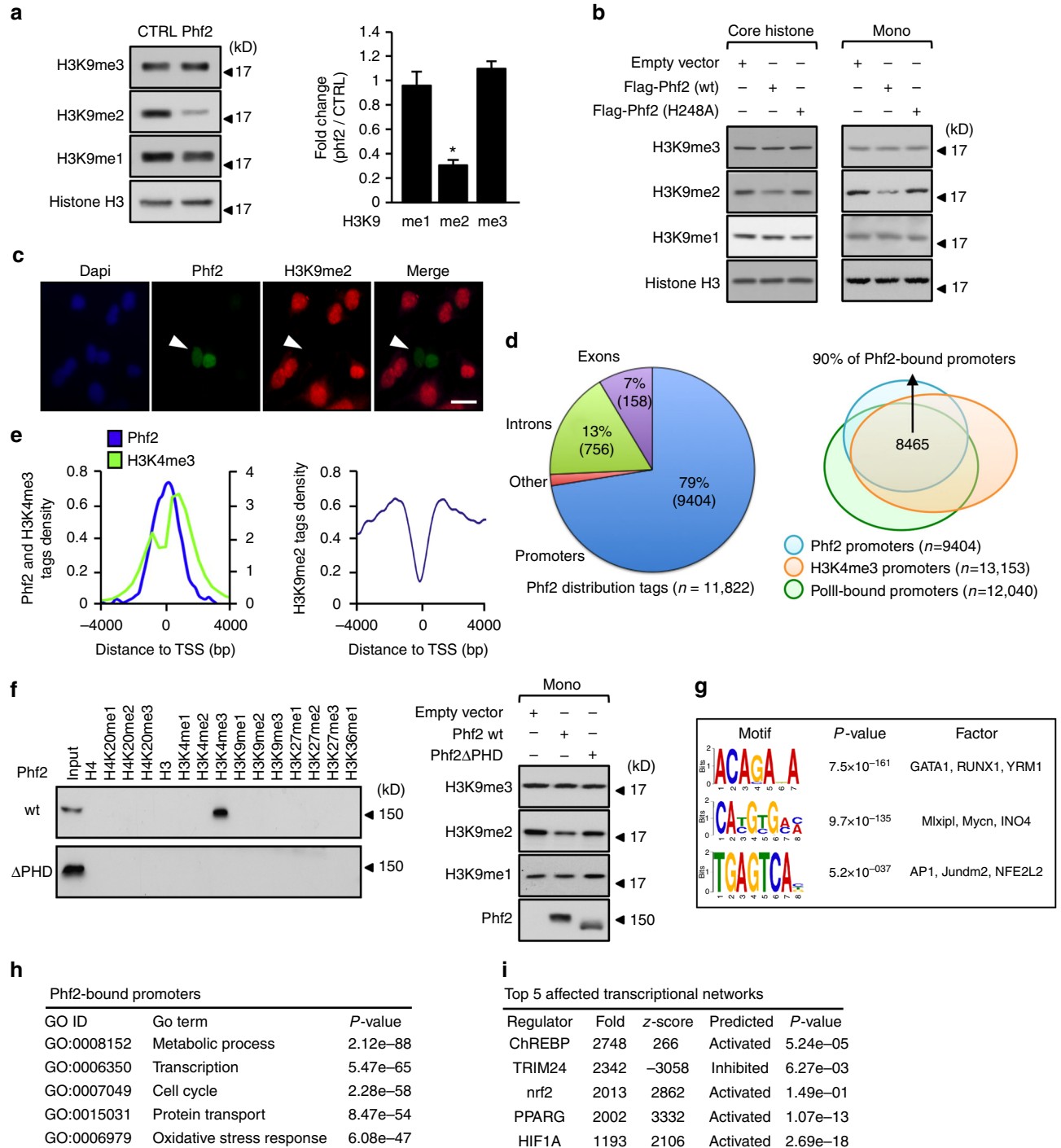

**Fig. 1** Characterization of Phf2 histone demethylase activity. **a, b** Demethylase activity of either FLAG-tagged wild-type (wt) or H248A mutant of Phf2 immunoprecipitated (IP) from 293T cell lysates, was assessed using either recombinant methylated histones (**a**), core histones or mononucleosomes (mono) (**b**) as substrates (n = 3). For panel **a**, a densitometric analysis of H3K9 mono, di or trimethylation levels normalized to histone H3 levels was performed, values are expressed as fold over the means of the CTRL transfected cells. **c** Primary hepatocytes were transfected with FLAG-tagged Phf2 expression vector for 24 h. Cells were then fixed and immunostained with the indicated antibody. Representative images shown (n = 3). Scale bars = 40 μm. **d** Genomic distribution of Phf2 and Venn diagrams showing overlap between Phf2-bound, RNA polII-bound and H3K4me3-marked promoters after ChIP-sequencing experiment from primary cultured hepatocytes incubated with 100 nM insulin and 25 mM glucose for 24 h. **e** Tag density plots displaying Phf2, H3K4me3, or H3K9me2 tags distribution relative to the transcriptional start site (TSS). **f** (Left panel) Peptide pull-down assay, mixing purified FLAG-tagged wt or ΔPHD Phf2 from 293T cells lysates with biotinylated histones tails. Pull-downs were analyzed by immunoblotting. (Right panel) Phf2 demethylase activity assessed using mononucleosomes as substrates (n = 4). **g** Top enriched motifs of Phf2 ChIP-seq peaks (n = 11,882). **h** Gene ontology analysis of Phf2-bound promoters from Phf2 ChIP-seq analysis. **i** Top enriched affected transcriptional networks in the liver of Phf2 overexpressing mice. Phf2 was overexpressed through adenoviral gene delivery in the liver of C57BL/6 J mice for 3 weeks. Microarray analysis was then performed to identify affected transcriptional networks. **a** Error bars represent mean ± SEM, *P < 0.01 Phf2 compared to CTRL (unpaired t-test)

hepatic steatosis development as revealed by increased liver tri-glyceride (TG), diacylglycerol (DAG), and cholesterol ester contents (Supplementary Table 1).

Despite severe lipid deposition, no modification in PKCε activity (Fig. 2b), and subcellular localization (Supplementary

Fig. 3d) was observed despite increased DAG content at both the plasma membrane and in the cytosol (Fig. 2c). Furthermore, total ceramide content, which can also trigger hepatocyte's PKCε activation, was not altered in liver of Phf2 mice compared to GFP (Supplementary Table. 1). In addition, serum levels of alanine

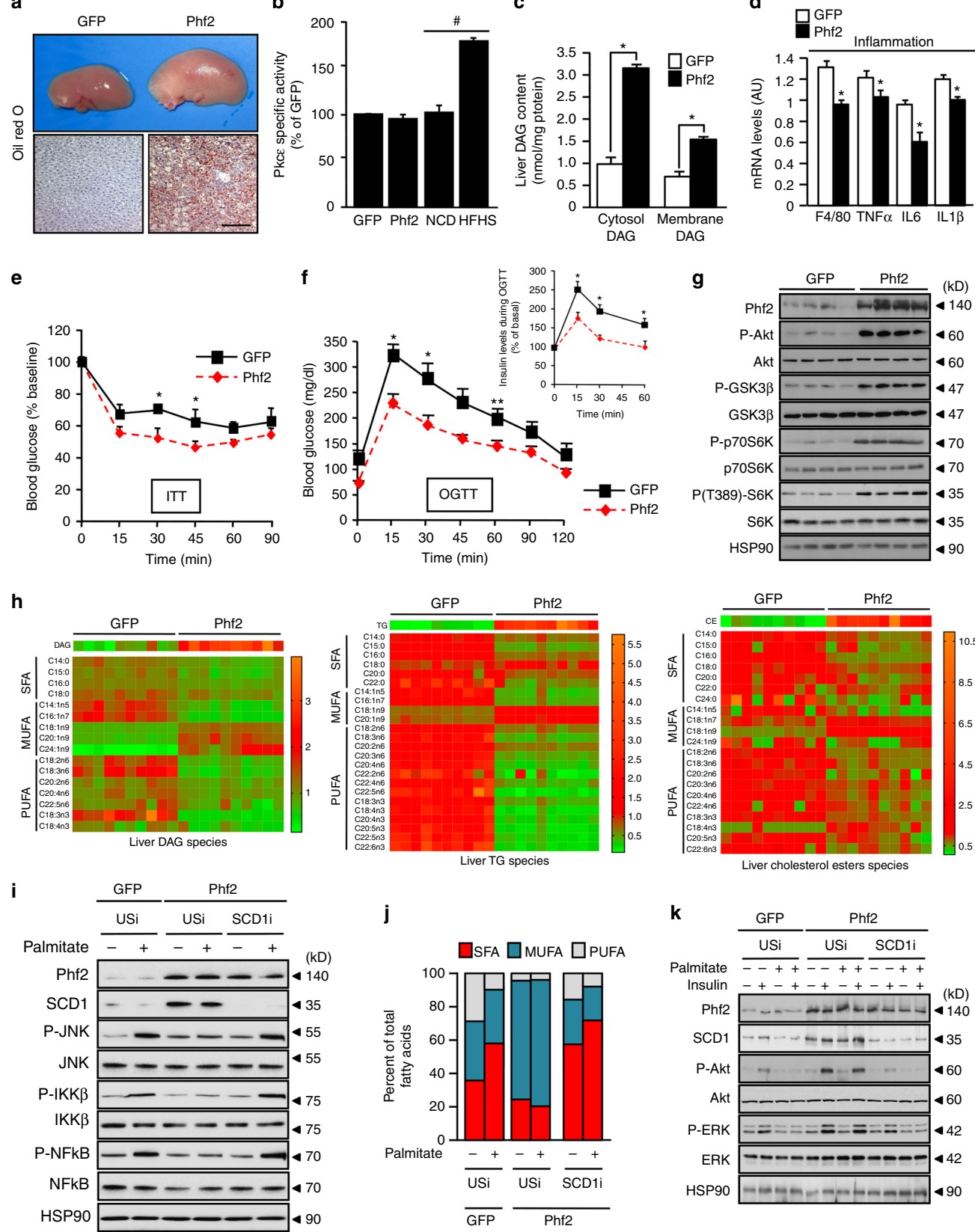

aminotransferase (ALAT) and aspartate aminotransferase (ASAT), which sign liver injury, were not modified (Supplementary Table 1). Macrophage infiltration (Supplementary Fig. 3e), and expression of pro-inflammatory cytokines such as Tnfα, IL6, and IL1β were even decreased Fig. 2d). Phf2 mice had lower post-prandial blood glucose and insulin levels (Supplementary Table 1). They were also more sensitive to insulin (ITT), more tolerant to glucose (OGTT) than controls, and exhibited decreased serum insulin levels during the glycemic burst (Fig. 2e, f). Pyruvate tolerance test (PTT) also revealed that insulin-mediated suppression of hepatic glucose production was potentiated upon Phf2 overexpression (Supplementary Fig. 3f). Furthermore, Phf2 mice showed enhanced liver PI3K/Akt signaling as evidenced by increased Akt, p70S6K, or GSK3β phosphorylation (Fig. 2g). However, no modification of the PI3K/Akt signaling was observed in the epididymal white adipose tissue (WAT) and skeletal muscle of liver-specific Phf2 over-expressing mice after insulin stimulation, suggesting absence of change in peripheral insulin sensitivity (Supplementary Fig. 3g). Supporting this conclusion, glycogen content in skeletal muscles was not altered after liver-specific Phf2 overexpression (Supplementary Table 1). Overall, this demonstrates that Phf2-mediated hepatosteatosis development is associated with exacerbated liver insulin sensitivity and is dissociated from inflammation.

To further identify metabolic pathways controlling the observed tolerance to lipotoxicity, the impact of Phf2 over-expression on liver lipid composition was examined by lipidomic analysis. Strikingly, liver of Phf2 mice contained elevated levels of mono-unsaturated FA (MUFAs) and significantly less saturated FA (SFA) esterified into all of lipid species tested such as DAG, TG, cholesterol esters (Fig. 2h), and even ceramides (Supplementary Fig. 3h). Overall, as lipotoxicity is generally attributed to SFAs, their conversion into specific MUFAs could be instrumental in the protective effect of Phf2 from inflammation and insulin resistance, by decreasing their intracellular concentrations. This implicated the desaturation of SFA, a limiting step catalyzed by the family of stearoyl-CoA desaturase (SCDs), as a potential mechanism underlying the phf2-driven lipid compositional changes. Consistently, SCD1 expression (Supplementary Fig. 3a, b) and activity (Supplementary Fig. 3i) were enhanced in phf2 mice. To further examine the impact of SCD1-mediated SFA desatura-tion in the resistance to lipotoxicity, SCD1 expression was inhibited in primary cultured hepatocytes. As previously observed in vivo, Phf2 overexpression increased SCD1 expression and favored the conversion of SFA into MUFA (Fig. 2i). Consequently the percentage of SFA content significantly decreased after Phf2 overexpression in all lipid species (Fig. 2j). As a result, Phf2 overexpression, although increasing both DAG and TG content (Supplementary Fig. 3j), protected hepatocytes from palmitate-induced insulin resistance and inflammation (Fig. 2i, k and Supplementary Fig. 3k). In contrast, SCD1 silencing (Fig. 2i), without affecting DAG or TG content (Supplementary Fig. 3j), abolished Phf2-driven SFA desaturation,

as revealed by increased percentage of SFA and reduced MUFA content in hepatocyte (Fig. 2j). As a result, sensitivity to palmitate-induced inflammation and insulin resistance were restored in Phf2 overexpressing hepatocytes (Fig. 2i, k). Altogether, our results suggest that Phf2-mediated SFA desaturation into DAG and TG protects the liver from inflammation and insulin resistance despite hepatic steatosis development.

**Phf2 diverts glucose fluxes to protect liver from oxidative damages**. Metabolic tracing studies further demonstrated that hepatic glucose and FA oxidation rates were enhanced in Phf2 mice (Fig. 3a). As a consequence, hepatic glutamate and succinate-driven mitochondrial respiration was increased in Phf2 mice, indicating higher mitochondrial oxidative capacity (Fig. 3b). Intriguingly, although these alterations could lead to ROS overproduction, since more substrate-derived electrons are entering the mitochondrial respiration chain, Phf2 mice did not show increase in ROS levels, protein carbonylation or proteasome activity (Fig. 3c and Supplementary Fig. 4a). As lipotoxicity, through ROS production, is one of the major events involved in NAFLD progression[24], these results further suggest that Phf2 activation, by improving oxidative stress defenses, could protect liver from the adverse consequences of lipid deposition (Supplementary Fig. 4b). Supporting an anti-oxidative stress response, metabolomic KEGG pathway analysis revealed that the pentose phosphate pathway shunt (PPP), serine, glycine, and glutathione biosynthesis showed a strong enrichment in Phf2 mice (Fig. 3d, e and Supplementary Fig. 4c). Accordingly, increased expression of glucose 6-phosphate dehydrogenase (G6PDH) and transketolase (TKT) further demonstrated that glucose flux is consumed to feed into the PPP, providing NADPH to sustain ROS-scavenger glutathione peroxidases (Gpx) activity (Fig. 3f–h). In addition, enhanced expression of phosphoglycerate dehydrogenase (phgdh), glutamate–cysteine ligase (GCLC), and glutathione synthetase (GSS) diverts glycolytic carbons (3-phosphoglycerate) into the serine, glycine, and glutathione (GSH) biosynthetic pathways to sustain GSH synthesis (Fig. 3i, Supplementary Fig. 4c and Supplementary Table 1). Therefore, Phf2 contributes, in addition to its role in lipid partitioning, to the protective effect against oxidative stress, by rerouting glycolytic flux and by regulating the activity of ROS detoxifying enzymes.

**Phf2 acts as a new ChREBP epigenetic activator**. In keeping with the observation that Phf2 interacts with ChREBP (Supplementary Fig. 1), bioinformatic analyses highlighted that Phf2 overexpression affected ChREBP transcriptional networks. Indeed, a significant proportion of ChREBP target genes (49.94%) were increased in response to Phf2 overexpression (Supplementary Fig. 5a, b). Furthermore, ChREBP ChIP-seq indicated that 74% of ChREBP-bound promoters corresponded to those binding Phf2, and that ChREBP and Phf2 genomic distribution covered the TSS of their common target gene promoter (Fig. 4a and

**Fig. 2** Liver-specific Phf2 overexpression causes quick-onset hepatosteatosis. **a–h** Mice, injected with either GFP or Phf2 overexpressing adenovirus, were studied 3 weeks later in the fed state. **a** Phf2 mice develop hepatic steatosis as shown by increased liver size and by oil red O staining of liver sections. Scale bars = 100 μm (n = 10 per group). **b** Measurement of PKCε activity (n = 6 per group). **c** Liver DAG content in the cytosol and at the plasma membrane (n = 8 per group). **d** Relative expression of liver pro-inflammatory genes (n = 10 per group). **e** Insulin tolerance test (n = 10 per group). **f** Oral glucose tolerance test and insulin levels during the OGTT test (n = 10 per group). **g** Western blot analysis of the PI3K/Akt signaling pathway in liver (n = 20 per group). **h** Heatmap visualization of relative SFA, MUFA, and PUFA content in liver DAG, TG, and cholesterol ester species (n = 10 per group). **i–k** Isolated primary hepatocytes overexpressing Phf2 and in which SCD1 expression was inhibited were incubated in the presence of palmitate (480 μM) for 24 h. **i** Representative western blots of the pro-inflammatory signaling pathway (n = 4). **j** Percentage of hepatocyte SFA, MUFA and PUFA content in indicated culture conditions (n = 3). **k** Representative western blot analysis of the PI3K/Akt signaling (n = 4). All error bars represent mean ± SEM. Statistical analyses were made using unpaired t-test. *P < 0.01 GFP compared to Phf2, **P < 0.05 GFP compared to Phf2. #P < 0.01 NCD compared to HFHS

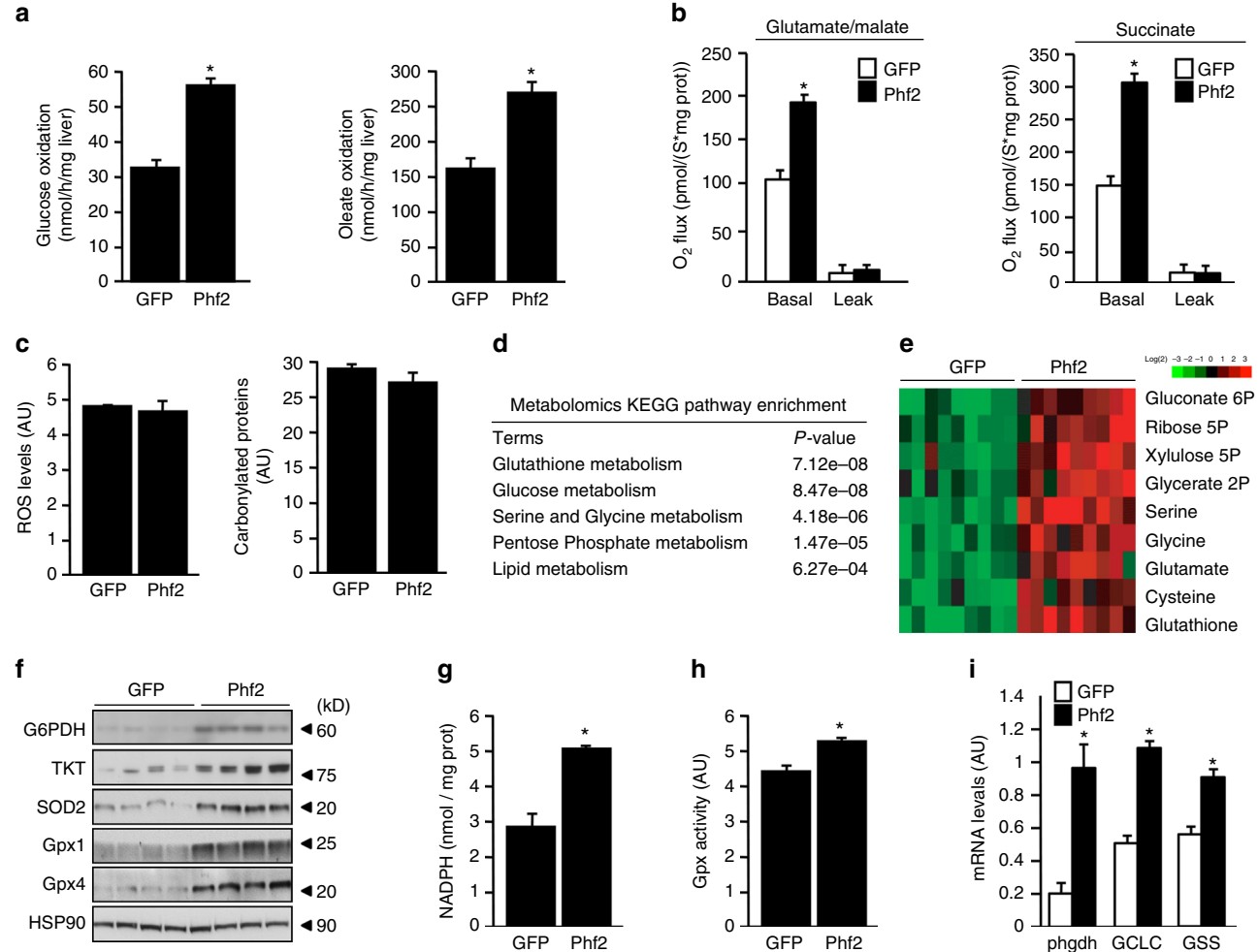

**Fig. 3** Phf2 diverts glucose fluxes to protect liver from oxidative stress. Mice, injected with either GFP or Phf2 overexpressing adenovirus, were studied 3 weeks later in the fed state. **a** Glucose and oleate oxidation rate determined by measuring the production of $^{14}CO_2$ from $^{14}C$-glucose or $^{14}C$-oleate for 4 h ($n = 8$ mice per group). **b** Basal glutamate/malate and succinate-driven mitochondrial respiration, which provide respectively electrons to the complex I and II of the mitochondrial reaction chain ($n = 6$ mice per group). **c** Relative ROS and carbonylated protein levels ($n = 12$ mice per group). **d** Metabolomic KEGG pathway enrichment analysis ($n = 15$ mice per group). **e** Heat map of metabolic intermediates of the PPP, serine, glycine, and GSH biosynthetic pathways ($n = 15$ mice per group). **f** Western blot analysis of proteins involved in oxidative stress defenses ($n = 12$ mice per group). **g** Liver NADPH content ($n = 10$ mice per group). **h** Measurement of Gpx activity ($n = 10$ mice per group). **i** Expression of genes involved in GSH synthesis ($n = 10$ mice per group). All error bars represent mean ± SEM. Statistical analyses were made using unpaired $t$-test. *$P < 0.01$ GFP compared to Phf2

Supplementary Fig. 5c). Accordingly, ChIP experiments confirmed that Phf2 is co-recruited with ChREBP on glycolytic and lipogenic gene promoter (Fig. 4b). Altogether, this suggests that Phf2 may interact with ChREBP for recruitment to the transcriptional complex. Supporting this hypothesis, Phf2 is recruited to the promoter of ChREBP-regulated genes in response to glucose stimulation, whereas its binding is significantly reduced after ChREBP silencing (Fig. 4c). These results demonstrate that the binding of Phf2 on the ChoRE-containing promoter is dependent on ChREBP. Consistent with a potential ChREBP coactivator function, Phf2 overexpression increased LPK and SCD1 promoter activity in vivo (Fig. 4d). In contrast, Phf2 silencing inhibited glycolytic and lipogenic gene expression and consequently decreased DNL and TG content (Supplementary Fig. 5d, e). At the chromatin level, in response to glucose stimulation, Phf2 silencing abolished H3K9me2 demethylation at the SCD1 promoter (Fig. 4e). Consistently, SCD1 chromatin promoter accessibility was decreased and the recruitment of ChREBP and the RNA polII were impaired (Fig. 4e). Collectively, this suggests that Phf2 contributes to the regulation of ChREBP

function by erasing H3K9me2 methyl-marks at the promoter of its target genes to increase transcription. Supporting this conclusion, ChREBP silencing abolished Phf2 action by reducing DNL and TG content in hepatocytes (Supplementary Fig. 5f, g).

Interestingly, there are now several examples where JmjC domain-containing histone demethylases with previously defined roles in histone demethylation also appear to demethylate non-histone proteins to regulate their abundance, stability or activity[25]. This realization that JmjC domain-containing demethylases potentially play widespread roles in protein demethylation raises an important question of whether Phf2 primary biological functions, in the regulation of ChREBP activity, is currently attributed to demethylase reactions toward histones or other uncharacterized non-histone proteins. To answer this question, a W29A mutant of Phf2 was overexpressed in cultured hepatocytes. This W29A mutation, localized within Phf2's PHD domain, has been previously shown to abolish H3K4me3 binding of Phf2 to the promoter of its target genes[26]. Accordingly, ChIP experiments, performed in cultured

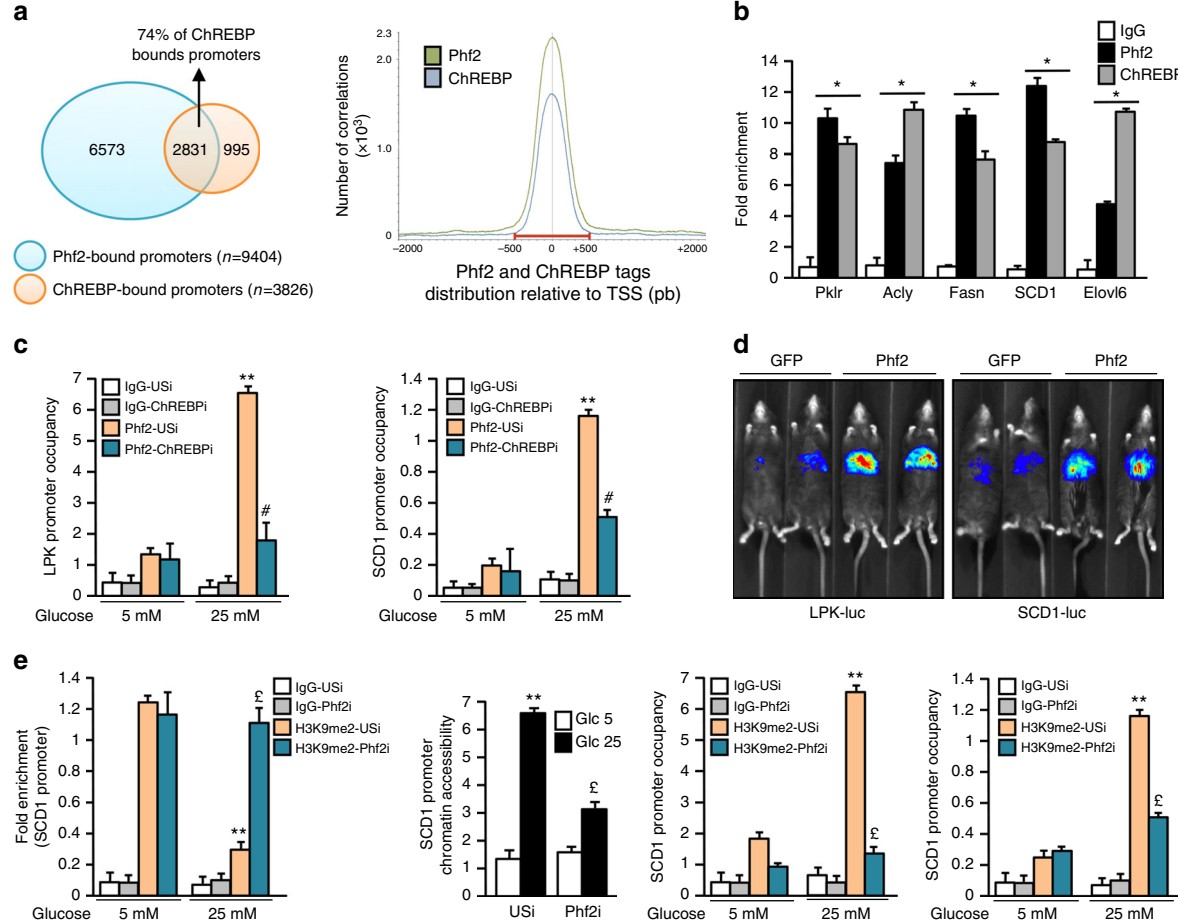

**Fig. 4** Phf2 acts as a new ChREBP transcriptional co-activator in hepatocytes. **a** Venn diagram of overlap between Phf2 and ChREBP ChIP-seq peaks on promoters in primary hepatocytes (left panel). Distribution of Phf2 and ChREBP ChIP-seq tags density in the vicinity of the TSS (right panel). **b** In vivo Phf2 and ChREBP ChIP experiments at the ChoRE-containing region of glycolytic and lipogenic gene promoters in the liver of C57Bl/6J fed mice ($n = 6$ per group). **c** ChIP experiments evaluating the recruitment of Phf2 to the LPK and SCD1 promoter after ChREBP silencing. Primary cultured hepatocytes, in which ChREBP expression was inhibited (ChREBPi), were incubated 24 h with either 5 or 25 mM glucose ($n = 5$). **d** Live imaging of LPK (*LPK-Luc*) and SCD1 (*SCD1-Luc*) promoter luciferase activity in the liver of fed mice overexpressing Phf2 ($n = 10$ per group). **e** Chromatin accessibility and ChIP experiments for H3K9me2, ChREBP, and RNA polII at the SCD1 promoter after Phf2 silencing. Primary cultured hepatocytes, in which Phf2 expression was inhibited (Phf2i), were incubated 24 h with either 5 or 25 mM glucose ($n = 3$). All error bars represent mean ± SEM. Statistical analyses were made using unpaired *t*-test (**b**) or Anova, followed by Bonferonni's test (**c**, **e**). *$P < 0.01$ compared to IgG, **$P < 0.01$ USi 5 mM compared to USi 25 mM, #$P < 0.01$ USi 25 mM compared to ChREBPi 25 mM, £ $P < 0.01$ USi 25 mM compared to Phf2i 25 mM

hepatocytes, confirmed that Phf2 W29A is no longer recruited on the promoter of glycolytic and lipogenic genes compared with WT Phf2 (Supplementary Fig. 6a). At the chromatin level, H3K9me2 demethylation at the SCD1 promoter was not increased by Phf2 W29A overexpression compared to WT Phf2 (Supplementary Fig. 6b). Consistently, both chromatin accessibility and ChREBP recruitment at the SCD1 promoter were not increased by Phf2 W29A overexpression (Supplementary Fig. 6c, d). As a consequence, Phf2 W29A is unable to enhance the expression of glycolytic and lipogenic genes and increase hepatocyte TG content (Supplementary Fig. 6e, f). The fact that Phf2 W29A conserved its histone demethylase activity toward recombinant proteins (Supplementary Fig. 6g), demonstrates that Phf2 contributes to the regulation of ChREBP function by erasing H3K9me2 methyl-marks at the promoter of ChREBP target genes, ruling out non-histone protein demethylation in this process. Supporting this conclusion, a H248A Phf2 mutant, with no histone demethylase activity (Fig. 1b), is also unable to stimulate ChREBP transcriptional activity in hepatocytes despite being functionally recruited on the promoter of ChREBP-regulated genes (Supplementary Fig. 6h–j).

**Phf2 and ChREBP regulate Nrf2 activity to enhance oxidative stress defenses**. In the search for the mediator of Phf2 action against oxidative stress, bioinformatic analysis revealed that Nrf2 transcriptional network, which plays a central role in the defense against oxidative stress[27], was significantly affected by Phf2 overexpression (Fig. 1h, i). Accordingly, Nrf2 protein content were increased in the liver of Phf2 mice (Fig. 5a). Furthermore, the activity of a Nrf2-reporter construct (ARE-luc) (Supplemental Fig. 7a) and the expression of specific Nrf2-regulated genes (p62 or NQO1) were enhanced upon Phf2 overexpression (Fig. 5a). In this context, our data demonstrated that Nrf2 was the major mediator of Phf2-beneficial effects, since its silencing, by reducing the expression of ROS scavenger proteins, reduced Phf2-mediated NADPH and GSH biosynthesis and, as a consequence, restored palmitate-induced ROS production and apoptosis (Fig. 5b–d). At the chromatin level, ChIP-seq analysis showed that Phf2 and ChREBP are functionally co-recruited to the promoter of Nrf2 in response to glucose stimulation (Fig. 5e). Supporting their role in controlling Nrf2 activity, Phf2 silencing reduced H3K9me2 demethylation at the Nrf2 promoter (Fig. 5f). Consistently, Nrf2 chromatin promoter accessibility and the recruitment of ChREBP

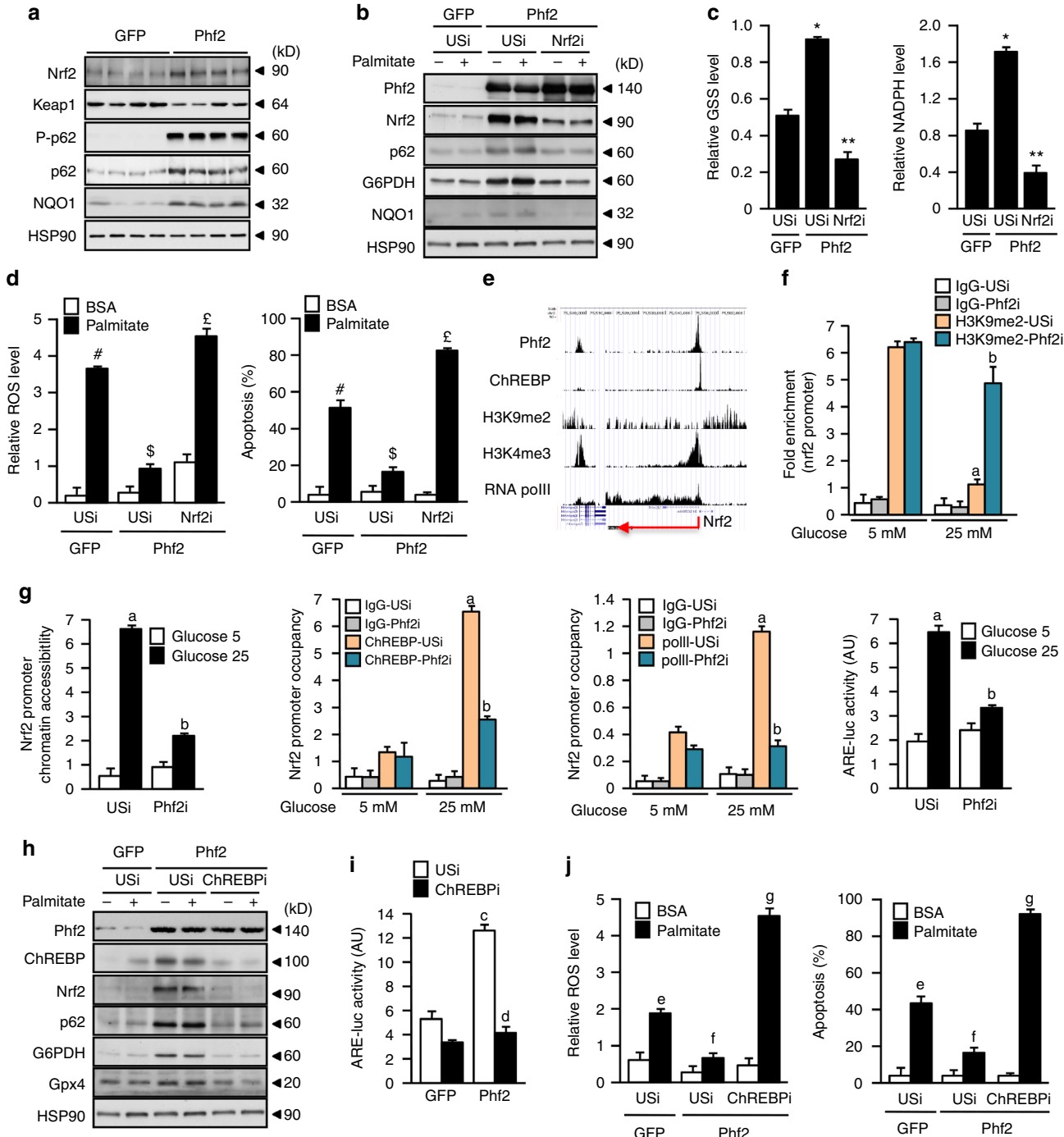

**Fig. 5** Phf2 and ChREBP decrease oxidative stress through Nrf2 activation. Mice, injected with either GFP or Phf2 overexpressing adenovirus, were studied 3 weeks later in fed state. **a** Representative Western blot analysis of Nrf2 protein content and Nrf2-regulated genes ($n = 10$ mice per group). **b–d** Nrf2 was inhibited in cultured hepatocytes overexpressing Phf2. **b** Western blot analysis of proteins involved in oxidative stress defenses ($n = 3$). GSH and NADPH contents (**c**) in addition to ROS levels and hepatocyte apoptosis (**d**) were determined after incubation with 480 μM palmitate for 24 h ($n = 3$). **e** UCSC genome browser image illustrating normalized tag counts for Phf2, ChREBP, H3K9me2, H3K4me3, and RNA polII at the Nrf2 promoter. (**f, g**) Phf2 was inhibited in cultured hepatocytes. ChIP for H3K9me2, ChREBP, and RNA polII, in addition to chromatin accessibility at the Nrf2 promoter and Nrf2 activity on the ARE-luc construct shown ($n = 3$). **h–j** ChREBP expression was inhibited in cultured hepatocytes overexpressing Phf2. **h** Representative western blot analysis showing the contribution of ChREBP to the regulation of Nrf2 and Nrf2-regulated gene expression. **i** Nrf2 activity on the ARE-luc construct shown. **j** Relative ROS levels and measurement of hepatocyte apoptosis were determined after incubation with 480 μM palmitate for 24 h ($n = 3$). All error bars represent mean ± SEM. Statistical analyses were made using Anova, followed by Bonferonni's test. *$P < 0.01$ Phf2/USi compared to GFP/USi, **$P < 0.01$ Phf2/Nrf2i compared to Phf2/USi, #$P < 0.01$ GFP/USi palmitate compared to GFP/USi BSA, $$P < 0.01$ Phf2/USi palmitate compared to GFP/USi palmitate, £$P < 0.01$ Phf2/Nrf2i palmitate compared to Phf2/USi palmitate, **a** $P < 0.01$ USi 25 mM glucose compared to USi 5 mM glucose, **b** $P < 0.01$ Phf2i 25 mM glucose compared to USi 25 mM glucose, **c** $P < 0.01$ Phf2/USi compared to GFP/USi, **d** $P < 0.01$ Phf2/ChREBPi compared to Phf2/USi, **e** $P < 0.01$ GFP/USi palmitate compared to GFP/USi BSA, **f** $P < 0.01$ Phf2/USi palmitate compared to GFP/USi palmitate, and **g** $P < 0.01$ Phf2/ChREBPi palmitate compared to Phf2/USi palmitate

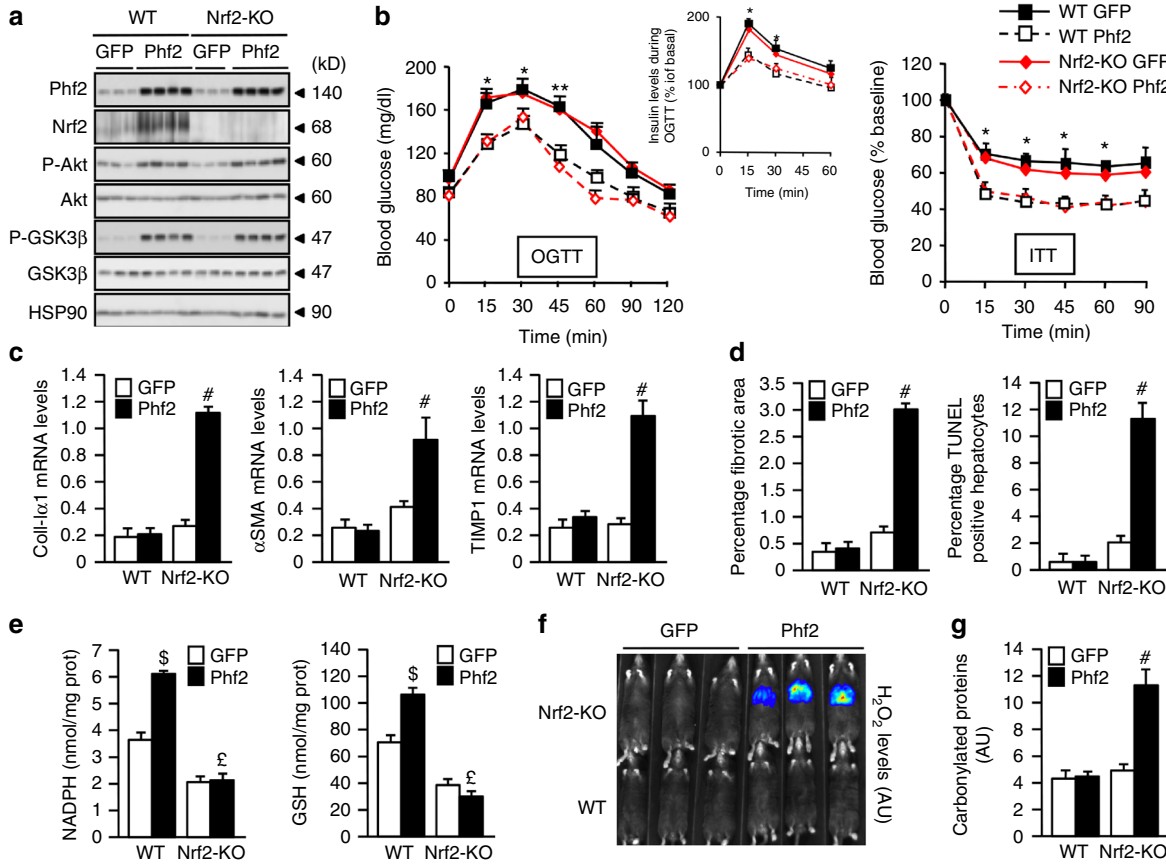

**Fig. 6** Phf2-driven Nrf2 activation protects liver from fibrogenesis. GFP or Phf2 were overexpressed in the liver of Wild Type (WT) and Nrf2 knockout (Nrf2-KO) mice. Mice were studied 3 weeks later in the fed state. **a** Western blot analysis of liver extracts from WT or Nrf2-KO mice ($n = 10$ mice per group). **b** (Left) Oral glucose tolerance test and insulin levels during the OGTT test. (right) Insulin tolerance test ($n = 10$ per group). **c** Expression of coll-Ia1, α-SMA, and TIMP-1 ($n = 10$ mice per group). **d** Percentage of fibrotic area and percentage of apoptotic hepatocytes ($n = 10$ per group). **e** NADPH and GSH contents ($n = 10$ mice per group). **f** In vivo bioluminescent response of the PCL-2 probe to $H_2O_2$. Representative image for mice injected with the PCL-2 probe is shown ($n = 6$ mice per group). **g** Levels of carbonylated proteins ($n = 10$ mice per group). All error bars represent mean ± SEM. Statistical analyses were made using unpaired $t$-test (**b**) or Anova, followed by Bonferonni's test (**c**–**e**, **g**). *$P < 0.01$ GFP compared to Phf2, **$P < 0.05$ GFP compared to Phf2, #$P < 0.01$ Nrf2 KO/GFP compared to Nrf2KO/Phf2, \$$P < 0.01$ WT/GFP compared to WT/Phf2, £$P < 0.01$ WT/Phf2 compared to Nrf2KO/Phf2

and the RNA polII were decreased, reducing Nrf2 transcriptional activity in response to glucose stimulation (Fig. 5g). In addition, ChREBP silencing abolished Phf2-driven Nrf2 activation and restored sensitivity to palmitate-induced ROS production and apoptosis in Phf2 overexpressing hepatocytes (Fig. 5h–j). Finally, these effects of Phf2, in regulating Nrf2 transcriptional activity, are also dependent on its H3K9me2 histone demethylase activity, since Phf2 H248A or W29A mutants are unable to enhance Nrf2 activity and protect hepatocytes from palmitate-induced ROS production and hepatocyte apoptosis (supplementary Fig. 7b–g).

**Phf2 protects the liver from fibrogenesis in an Nrf2-dependent manner.** To investigate the contribution of Nrf2 to the protective effects of Phf2 during NAFLD progression, Phf2 was overexpressed in the liver of wild type (WT) or Nrf2 knockout (Nrf2-KO) mice (Fig. 6a). Phf2 overexpression increased hepatic TG content and enhanced MUFA/SFA ratio in a similar manner regardless of the genotype (Supplementary Fig. 8a and Supplementary Table 2). No pro-inflammatory response was observed in liver of WT or Nrf2-KO mice in response to Phf2 overexpression (Supplementary Fig. 8b). Both WT and Nrf2-KO mice overexpressing Phf2 were also more tolerant to glucose (OGTT) compared to GFP and elicited decreased serum insulin levels during the glycemic burst (Fig. 6b). In addition, Phf2 overexpressing WT and Nrf2-KO mice were more sensitive to insulin

(ITT) compared to GFP mice in both genotypes (Fig. 6b). Hepatic insulin sensitivity was also improved to the same level in both Phf2 overexpressing WT and Nrf2-KO mice compared to GFP, as evidenced by increased Akt or GSK3β phosphorylation (Fig. 6a). However, enhanced expression of coll-Ia1, α-SMA or TIMP-1 demonstrated that Phf2 overexpression promoted the progression into fibrosis only in the context of Nrf2 deficiency (Fig. 6c and Supplementary Fig. 8c). Concomitantly, circulating levels of ALAT and ASAT, as well as the number of apoptotic cells and fibrotic areas were only increased in the liver of Nrf2-KO mice overexpressing Phf2 (Fig. 6d and Supplementary Table 2). In Nrf2-KO mice, Phf2 overexpression no longer induced anti-oxidative stress response compared to WT mice as evidence by reduced NADPH and GSH biosynthesis (Fig. 6e and Supplementary Fig. 8d). As it is generally accepted that ROS play a key role in chronic liver injury, their reduced detoxification, illustrated with elevated levels of liver hydrogen peroxide ($H_2O_2$) and protein carbonylation, is likely to contribute to the severe liver injury observed in Nrf2-KO mice overexpressing Phf2 (Fig. 6f, g). Overall, these results demonstrate that enhanced Nrf2 activity is determinant to prevent the progression into fibrosis in the context of Phf2-induced hepatic steatosis development.

**Phf2 protects mice from obesity, insulin resistance, and fibrogenesis.** To test whether deregulation of Phf2 activity occurs

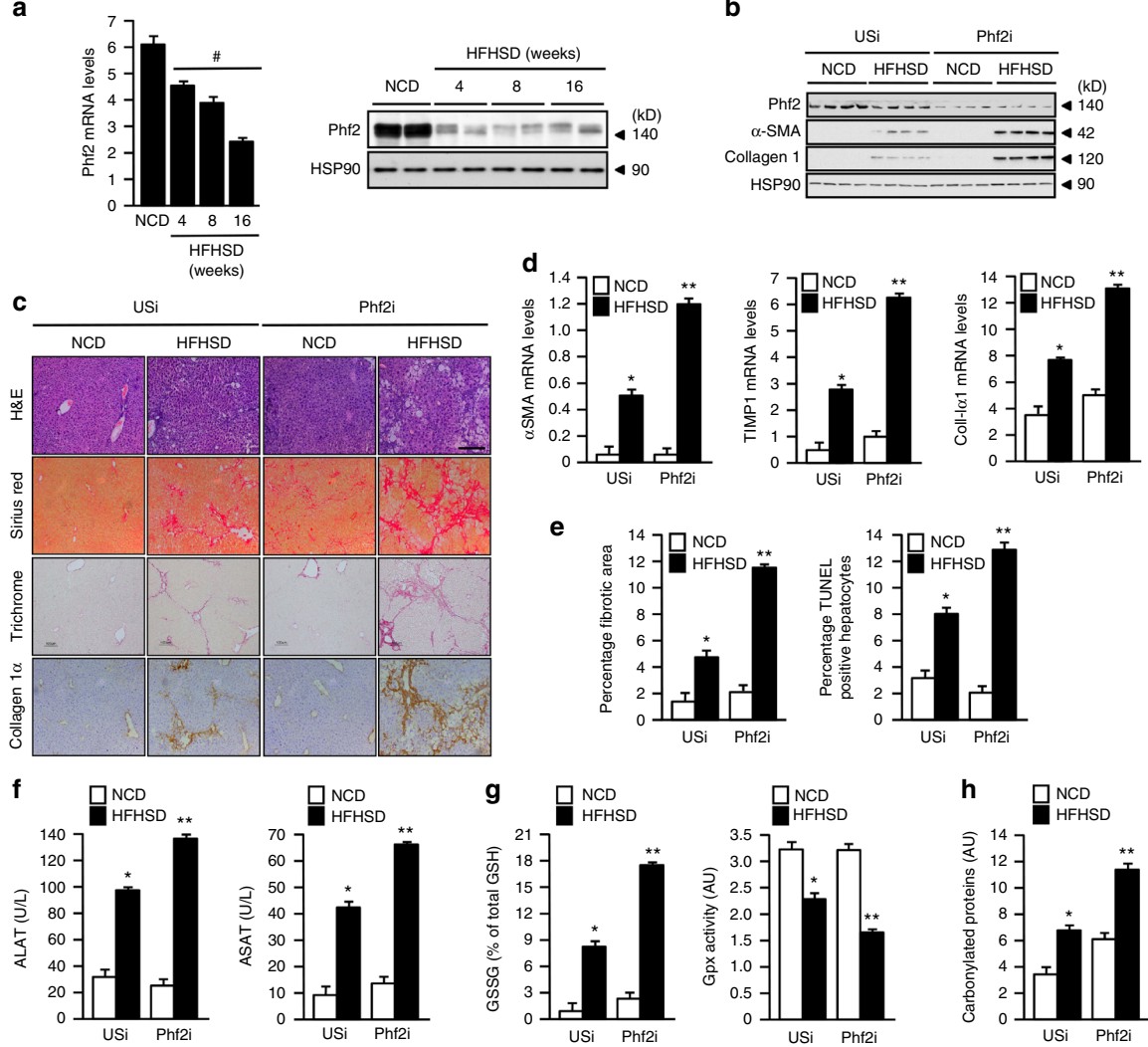

**Fig. 7** Phf2 silencing favors liver fibrosis development upon HFHSD feeding. **a** Relative expression at the mRNA and protein levels of Phf2 in the liver of C57Bl6/J mice trough the time course of HFHSD feeding ($n = 8$). **b–h** Phf2 expression was stably inhibited, through AAV strategy, specifically in the liver of C57Bl/6J mice fed with either a normal chow diet (NCD) or with a high fat and high sucrose diet (HFHSD) for 16 weeks. Mice were then studied in the fed state. **b** Representative western blot analysis of α-SMA and collagen I in the liver of USi or Phf2i mice ($n = 12$ per group). **c** Liver sections stained with hematoxylin and eosin (H&E), sirius red, trichrome masson and collagen Iα are shown. Scale bars = 100 μm ($n = 6$ per group). **d** Expression of coll-Ia1, α-SMA and TIMP-1 ($n = 12$ mice per group). **e** Percentage of fibrotic area and percentage of apoptotic hepatocytes ($n = 12$ per group). **f** Circulating levels of ALAT and ASAT ($n = 12$ per group). **g** Relative oxidized GSSG content and measurement of Gpx activity compared to USi ($n = 6$). **h** Levels of carbonylated proteins ($n = 6$ mice per group). All error bars represent mean ± SEM. Statistical analyses were made using Anova, followed by Bonferonni's test. $^{\#}P < 0.01$ HFHSD compared to NCD, $^{*}P < 0.01$ NCD/USi compared to HFHSD/USi, $^{**}P < 0.01$ HFHSD/Phf2i compared to HFHSD/USi

during NAFLD progression, its expression was measured during high fat and high sucrose diet-induced obesity (HFHSD). In this context, Phf2 expression was gradually decreased in the liver of mice fed a HFHSD diet when compared to mice fed on a normal chow diet (NCD) (Fig. 7a). To determine if decreased Phf2 activity could indeed favor the entry of NAFLD into fibrosis after HFHSD feeding, Phf2 expression was stably inhibited, specifically in the liver, through the use of an associated adenovirus (AAV) strategy (Fig. 7b). To achieve this specificity, an unspecific shRNA (USi) or Phf2 shRNA (Phf2i) were expressed under the control of the albumin promoter. As a result, Phf2 silencing enhanced liver fibrosis development when compared to USi mice upon HFHSD feeding (Fig. 7b, c). The expression of Col1a1, TIMP1, and a-SMA was also increased in liver of Phf2i mice compared to USi mice along with collagen deposition, number of fibrotic areas and hepatocyte apoptosis (Fig. 7c–e). In addition, serum levels of ALAT and ASAT were further increased in Phf2i mice (Fig. 7f).

Phf2 silencing also increases liver oxidative damages compared to USi mice facilitating the conversion of NAFLD into fibrosis upon HFHSD feeding (Fig. 7g, h).

Consequently, to determine whether Phf2 activation could protect liver from fibrosis development during obesity, Phf2 (FLAG-tagged) was stably overexpressed specifically in the liver, through a similar AAV strategy. To achieve hepatic specificity, GFP or FLAG-Phf2 expression was under the control of the albumin promoter (Fig. 8a). Compared to GFP, Phf2 mice were fully protected from HFHSD-induced obesity, despite the fact that detailed analysis of food intake, using lean body mass as a covariate, revealed that they were hyperphagic on both NCD and HFHSD (Supplementary Table 3). In addition, compared to HFHSD-GFP mice, which exhibited a pro-inflammatory response in the liver (increased Il-6, Il-1β, or Tnf-α expression), Phf2 mice were protected from inflammation (Supplementary Fig. 9a). In addition, Phf2 mice did not develop fasting hyperglycemia and

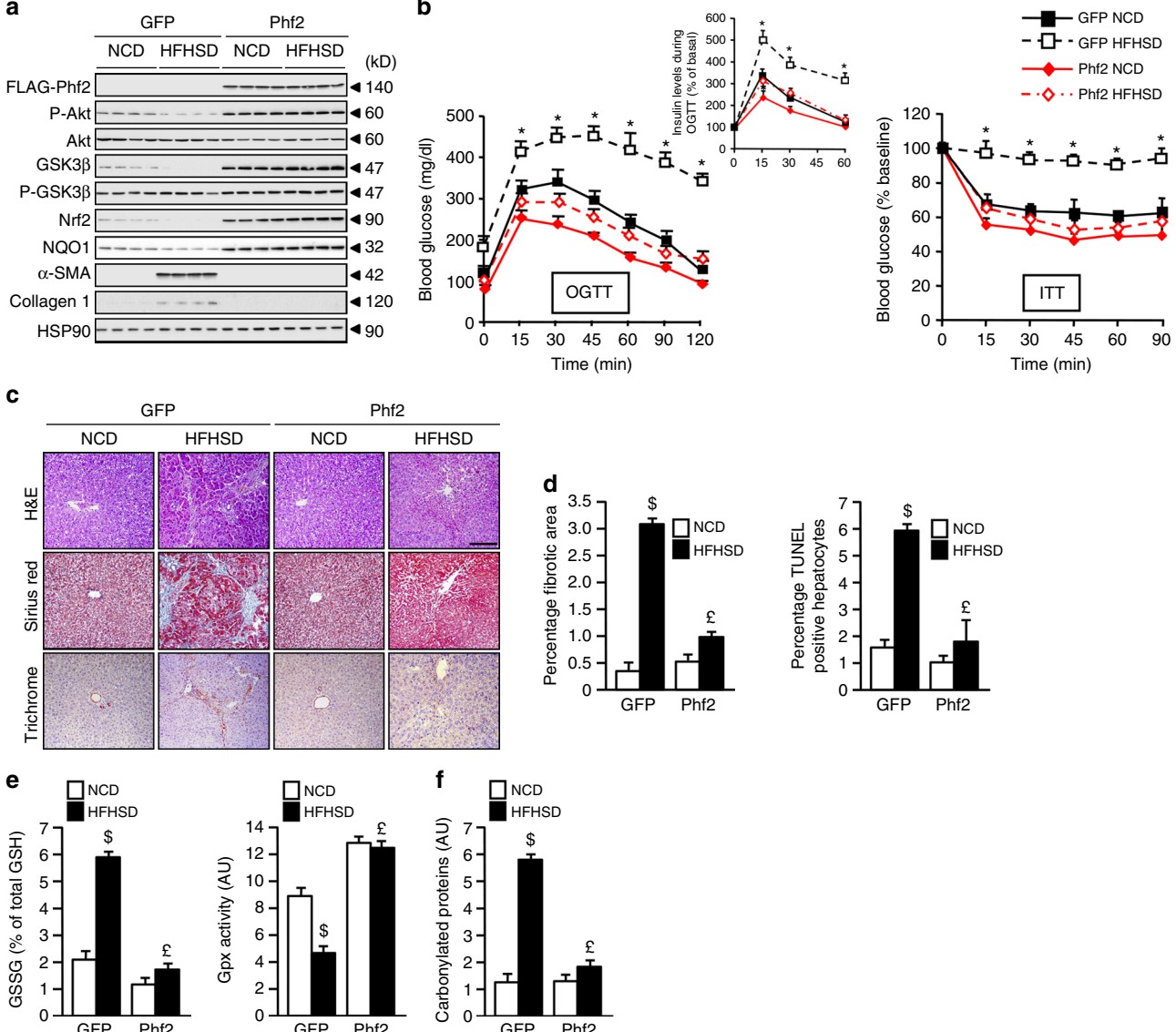

**Fig. 8** Phf2 protects the liver from high fat and high sucrose diet-induced fibrogenesis during obesity. GFP or Phf2 were overexpressed through AAV strategy specifically in the liver of 7-week-old male C57Bl/6 J mice. Mice were then fed with either a chow diet (NCD) or with a high fat and high sucrose diet (HFHSD) for 16 weeks. Mice were studied in the fed state. **a** Representative western blot analysis showing the contribution of Phf2 overexpression to the regulation of the PI3K/Akt signaling and pro-fibrogenic pathways in liver extracts (n = 15 per group). **b** (Left) Oral glucose tolerance test and insulin levels during the OGTT test. (right) Insulin tolerance test (n = 15 per group). **c** Liver sections stained with hematoxylin and eosin (H&E), trichrome masson and sirius red are shown. Scale bars = 100 μm (n = 10 per group). **d** Percentage of fibrotic area and percentage of apoptotic hepatocytes (n = 15 per group). **e** Relative oxidized GSSG content and measurement of Gpx activity (n = 10 per group). **f** Levels of carbonylated proteins (n = 10 mice per group). All error bars represent mean ± SEM. Statistical analyses were made using unpaired t-test (**b**) or Anova, followed by Bonferonni's test (**d, e**). *$P < 0.01$ HFHSD/GFP compared to HFHSD/Phf2, $^{\$}P < 0.01$ HFHSD/GFP compared to NCD/GFP, $^{£}P < 0.01$ HFHSD/Phf2 compared to HFHSD/GFP

fasting hyperinsulinemia as compared to HFHSD-GFP mice (Supplementary Table 3). Furthermore, OGTT and ITT tests confirmed that HFHSD Phf2 mice had an overall improved glucose tolerance and insulin sensitivity compared to GFP mice fed on the same diet (Fig. 8b). Interestingly, Phf2 mice were also protected from the development of insulin resistance in liver, WAT, and skeletal muscle upon HFHSD feeding compared to GFP mice, suggesting that the overall improvement of their insulin sensitivity results from sustain activation of the PI3K/Akt signaling in these tissues (Fig. 8a and Supplementary Fig. 9b). More importantly, liver histological analyses revealed that HFHSD Phf2 mice were protected against fibrogenesis compared to HFHSD GFP mice, in which collagen deposition (Fig. 7c), number of fibrotic area and hepatocyte apoptosis (Fig. 8d) were

increased along with the expression of Col1a1, TIMP1, and a-SMA (Supplementary Fig. 9c). Supporting these data, serum levels of ALAT and ASAT were not increased in Phf2 mice after HFHSD feeding (Supplementary Table 3). Overall, Phf2 over-expression, by increasing the activity of Nrf2 transcriptional network (Fig. 8a), enhanced oxidative stress defenses and consequently protected the liver from oxidative damages during HFHSD feeding (Fig. 8e, f).

**Phf2 expression is increased in the liver of obese patients with steatosis.** Finally, to evaluate the relevance of deregulated Phf2 activity during non alcoholic fatty liver (NAFL) progression in human, liver biopsies from lean subjects with no fatty liver

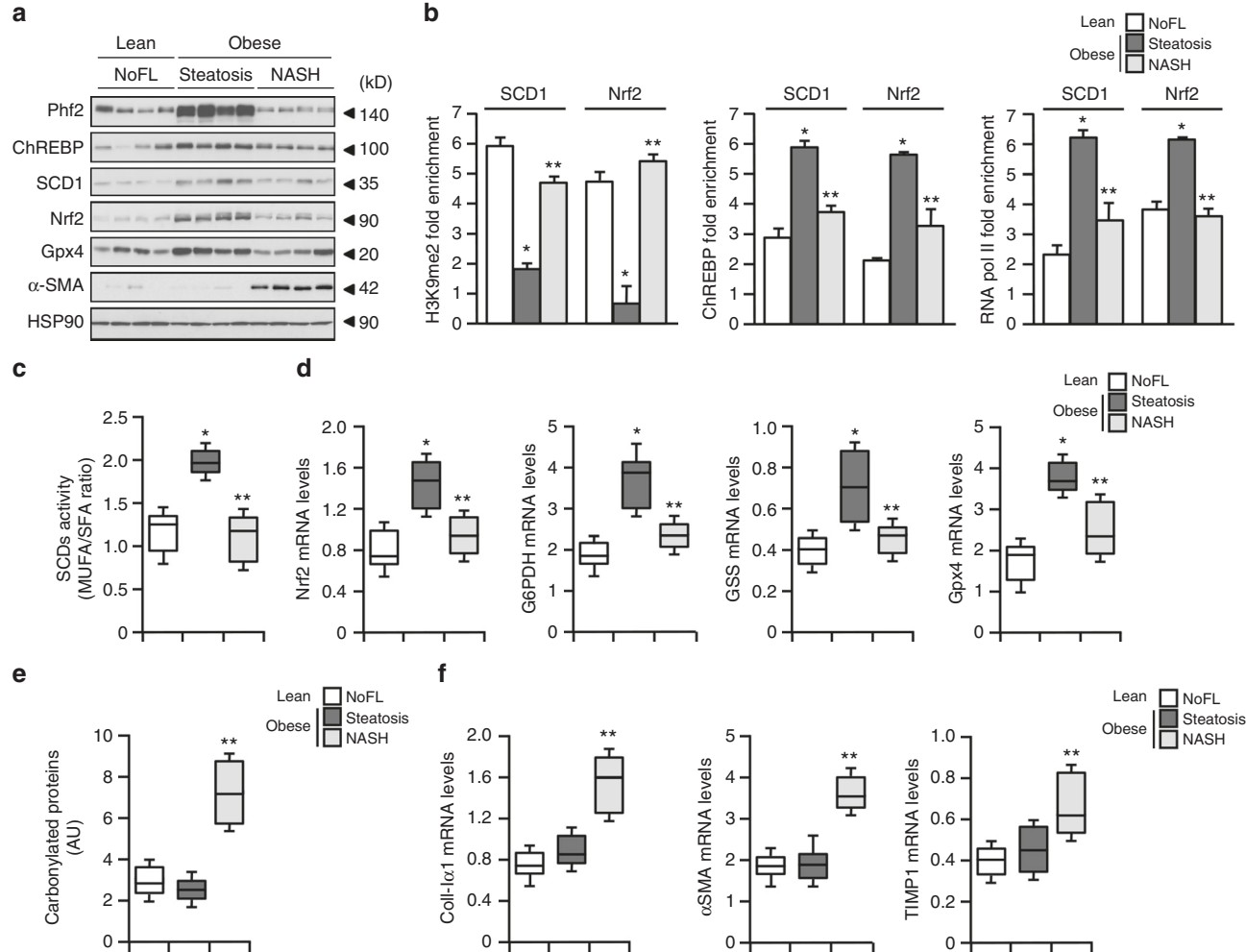

**Fig. 9** Phf2 expression is increased in the liver of obese patients with benign hepatic steatosis and positively correlates with insulin sensitivity. Human liver biopsies from lean subjects with no fatty liver (NoFL) or obese patients with simple steatosis or NASH were obtained from the ABOS cohort. **a** Representative Western blot analysis of Phf2, ChREBP, SCD1, Nrf2, Gpx4, and α-SMA expression ($n = 10$ per group). **b** ChIP experiments for H3K9me2 levels and for the recruitment of ChREBP and the RNA polII at the SCD1 and Nrf2 promoter ($n = 10$ per group). **c** MUFA/SFA ratio reflecting SCD1 activity ($n = 10$ per group). **d** Expression of Nrf2 and Nrf2-regulated genes ($n = 10$ per group). **e** Levels of carbonylated proteins ($n = 10$ per group). **f** Expression of coll-Ia1, α-SMA, and TIMP-1 ($n = 10$ per group). All error bars represent mean ± SEM. Statistical analyses were made using Anova, followed by Bonferonni's test. *Obese with steatosis compared to lean with noFL; $P < 0.01$. **Obese with steatosis compared to obese with NASH; $P < 0.05$

(NoFL) or obese patients with same BMI and same degree of hepatic steatosis (60%), were obtained from the Biological Atlas of Severe Obesity (ABOS) cohort (Supplementary Tables 4 and 5). The obese and steatotic cohort was further subdivided into two groups showing either simple steatosis or NASH based on their histological scoring. Compared to NoFL and NASH patients, Phf2 expression was only increased in the liver of insulin-sensitive steatotic patients (Fig. 9a). Correlating with Phf2 expression, H3K9me2 levels at the SCD1 and Nrf2 promoters were reduced in the liver of steatotic patients (Fig. 9b). Accordingly, ChREBP and RNA polII recruitment to these two promoters was enhanced (Fig. 9b). Therefore, SCD1 activity was increased in the liver of steatotic patients as evidence by enhanced MUFA/SFA ratio (Fig. 9c). Furthermore, Nrf2 and Nrf2-regulated gene expression was also potentiated (Fig. 9d). As a consequence, despite the same percentage of liver fat content, protein carbonylation was not increased in the liver of steatotic patients as compared to NASH patients (Fig. 9e). This apparent protection against oxidative stress in the liver of steatotic patients was correlated with reduced expression of pro-fibrogenic genes as compared to NASH patients (Fig. 9f). Overall, our data reveal in

human that Phf2 activation, and subsequent H3K9me2 demethylation at the promoter of ChREBP-regulated genes, occurs during hepatosteatosis development to protect the liver from oxidative stress and fibrosis development (Fig. 9g).

## Discussion

Emerging evidence that epigenetic processes convert alterations in metabolism into heritable pattern of gene expression has profound implications in understanding metabolic disorders[8,10,28]. Several studies have shown that NAFLD development and progression are correlated with changes in the pattern of histone methylation profiles. In this line of evidence, hepatic lipid accumulation leads to the aberrant histone H3K4 and H3K9 methylation in PPARα and lipid catabolism related genes, suggesting that histone methylation may contribute to hepatic steatosis and disease progression[29]. However, the exact nature of those epigenetic modifiers involved in the development and progression of the NAFLD spectrum remains partially unknown to date. Among all epigenetic modifiers, members of the KDM7 histone demethylase family, by removing repressive histone

methylation marks on the chromatin, are believed to act as transcriptional coactivators[26,30]. This suggests that their functions might be dependent on their selective association with specific transcription factors. Accordingly, our study unravels that Phf2 interacts with ChREBP and enhances ChREBP-driven transcription. Overall, Phf2-mediated H3K9me2 demethylation at ChREBP-regulated gene promoters favors DNL and TG accumulation in the liver. Supporting its metabolic function, Phf2 has been shown to stimulate fat storage in adipocytes, by coactivating FXR and CEBPα activity[17,19]. Moreover, mice with targeted disruption of Arid5b (AT-rich interactive domain D), a specific Phf2 coactivator partner[19,20], display a reduction of their white adipose tissue mass[21] as a result of reduced PPARγ activity [22]. Overall, our study supports that Phf2 and Arid5b work together in multiple organs as determinant regulators of lipid homeostasis. More importantly, while triggering hepatic TG accumulation, Phf2 concomitantly protects the liver from lipotoxicity by buffering the accumulation of detrimental FA. This reinforces the concept of lipoexpediency, in which optimizing lipid signals generated by DNL redirects fat toward benefit, even in the setting of lipid overload[31–34]. Our study particularly provides novel mechanistic insight into SCD1 action through the identification of Phf2 as a key target regulator of its expression and likely effector of its beneficial effect. Indeed, by modifying the MUFA/SFA balance in favor of MUFA synthesis, Phf2 decreases hepatic inflammation and insulin resistance. This supports studies showing that lipotoxicity, generally attributed to SFA, can be prevented by addition of MUFA[35–38]. In fact, according to this model, when present and stored in the proper location and time, specific lipid species may trigger signals that modulate adaptation to stress[34]. The effect of liver-specific Phf2 overexpression seems to be restricted to the liver, since no change in the PI3K/Akt signaling was detected in adipose tissue or skeletal muscle upon insulin stimulation. In this context, clamp study could be of interest to draw definitive conclusion in a follow up study regarding this restricted impact of liver-specific Phf2 overexpression on peripheral insulin sensitivity.

Our study further highlights the importance of Phf2 in regulating oxidative stress defenses. Central to these processes, Nrf2 is considered as one of the major transcription factors involved in the defense against oxidative stress[39–41]. In fact, Nrf2 expression is gradually down regulated in the end-stages of human liver diseases[42] and its deficiency in mice results in fibrosis development[43–45]. Therefore, the control of Nrf2 activity appears to be an important homeostatic mechanism that protects liver from nutrient-induced NAFLD progression[46,47]. Accordingly, our study specifically uncovers a new epigenetic mechanism for the control of Nrf2 activity, in which Phf2, by promoting H3K9me2 demethylation, enhances ChREBP-driven Nrf2 expression. In addition, and independently of the regulation of its expression, Nrf2 activity is also regulated through ubiquitination and subsequent degradation in a Keap1-dependent manner[48]. The autophagy-adapter p62 has been particularly shown to interact with the Nrf2-binding site of Keap1 and competitively inhibits the Keap1-Nrf2 interaction. In this setting, it has been described that Serine 351 (S351) of p62 is phosphorylated in a PI3K/ Akt /mTORC1-dependent fashion, causing p62's affinity for Keap1 to rise[49]. As a result, Nrf2 protein is stabilized and is then able to induce the expression of anti-oxidant enzymes[50,51]. Interestingly, our study further demonstrates that Phf2 overexpression by increasing mTORC1 activity (Fig. 2g) enhances S351 p62 phosphorylation and decreases Keap1 protein levels (Fig. 5a). Altogether, our results demonstrate that, in addition to its direct effect on Nrf2 expression, Phf2 overexpression, by stimulating the PI3K/Akt/mTORC1 signaling pathway may thus further participate in the stabilization and induction of Nrf2 and help the

expression of genes involved in the PPP, glutathione biosynthesis and anti-oxidative stress response.

Consistently, the down-regulation of Phf2 activity in the liver of NASH patients from the ABOS cohort positively correlates with low liver Nrf2 protein content and the high risk of progression into fibrosis. However, despite an active pro-fibrogenic response due to impaired anti-oxidant capacities, Nrf2 KO mice overexpressing Phf2 did not developed insulin-resistance as a result of enhanced oxidative stress. In the contrary, these mice are even more sensitive to insulin compared to Nrf2 KO mice overexpressing the GFP. These results are in agreement with the phenotype observed in Gpx1 KO mice, in which a decrease in ROS detoxification capacity enhances insulin sensitivity[52]. From this, one of the most important implications to our work is the concept that inflammation, insulin resistance and oxidative stress are prerequisite for fibrosis development should be reconsidered. This is somehow unexpected because it is commonly believed that oxidative stress is always linked to insulin resistance[46,53]. Through the use of Nrf2 KO mice, our results suggest that as long as the liver is able to manage the excess of lipids, by either converting them into MUFA and stored them into lipid droplets, it will be protected from inflammation and insulin resistance, despite the progression into fibrosis due to impaired ROS detoxification capacities. Consequently, we should not systematically consider fibrosis development as a direct consequence of inflammation and insulin resistance, which further reinforce the concept of multiple parallel hits hypothesis for explaining the NAFLD spectrum[3]. In human, when it comes to NAFLD progression, our study highlights the fact that lipid composition is critical for regulating inflammation and insulin sensitivity whereas anti-oxidative stress defenses, under the control of Nrf2 transcriptional activity, are crucial for controlling fibrosis development.

In conclusion, our findings establish in mice and human, a new epigenetic checkpoint, whereby Phf2 induction, through facilitating H3K9me2 demethylation at ChREBP regulated-gene promoters, protects liver from the accumulation of pathogenic lipids and ROS during NAFLD (Fig. 10). Taking into account Phf2 expression and activity as a reliable biomarker, our study could also lead to disease stratification, from simple steatosis to NASH and fibrosis, providing new tracks in NAFLD pathogenesis. In addition, since epigenetic modifications are reversible, novel therapies intended to modulate epigenetic abnormalities are trend today. In this line of evidence, given the availability of small molecules to specifically activate jmjC-containing histone demethylases[54–56], our study sheds light in the identification of Phf2 as potential "druggable" epigenetic target to prevent NAFLD progression.

## Methods

**Antibodies**. The following antibodies were used in this study for either western blotting, ChIP-sequencing, immunohistochemistry and chromatin immunoprecipitation. SOD2 (The Binding site, PC096), Gpx1 and Gpx4 (gift from L. Chavatte), ChREBP (Novus, NB400), SCD1 (Cell Signaling, C12H5), Phf2 (Cell Signaling, D45A2), H3K9me1 (Abcam, ab8896), H3K9me2 (Abcam, ab1220), H3K9me3 (Abcam, ab8898), histone H3 (Diagenode, C15200011), p(S473)-AKT (Cell signaling, D9E), Akt (Cell Signaling, 9272), P-p70S6K (Cell Signaling, 108D2), p70S6K (Cell Signaling, 49D7), p-GSK3β (Cell Signaling, D17D2), GSK3β (Cell Signaling, 27C10), LPK (Abcam, ab6191), ACC (Cell Signaling, 3662), FAS (Cell signaling, C20G5), p62 (gift from A. F. Burnol), Nrf2 (Santa Cruz Biotechnology, 13032), G6PDH (Cell Signaling, 12263), TKT (Cell Signaling, 8616), α-SMA (Novus, NB-600-531), collagen I (Novus, NB600-408), GAPDH (Santa Cruz Biotechnology, FL-335), malic enzyme (Novus, 68578), MTTP (Novus, 62489), NQO1 (Novus, A180), RNA PolII (Santa Cruz, sc899), HSP90 (Cell Signaling, 4874), Cidec (Abcam, ab77115), Plin2 (Progen, ap125), G0S2 (Santa Cruz Biotechnology, N13), and FLAG (Sigma, F1804).

**Plasmids and cloning procedures**. Full-length (WT) or truncated form (Phf2ΔPHD) of Phf2 was PCR-amplified from cDNA samples from 293T cells by

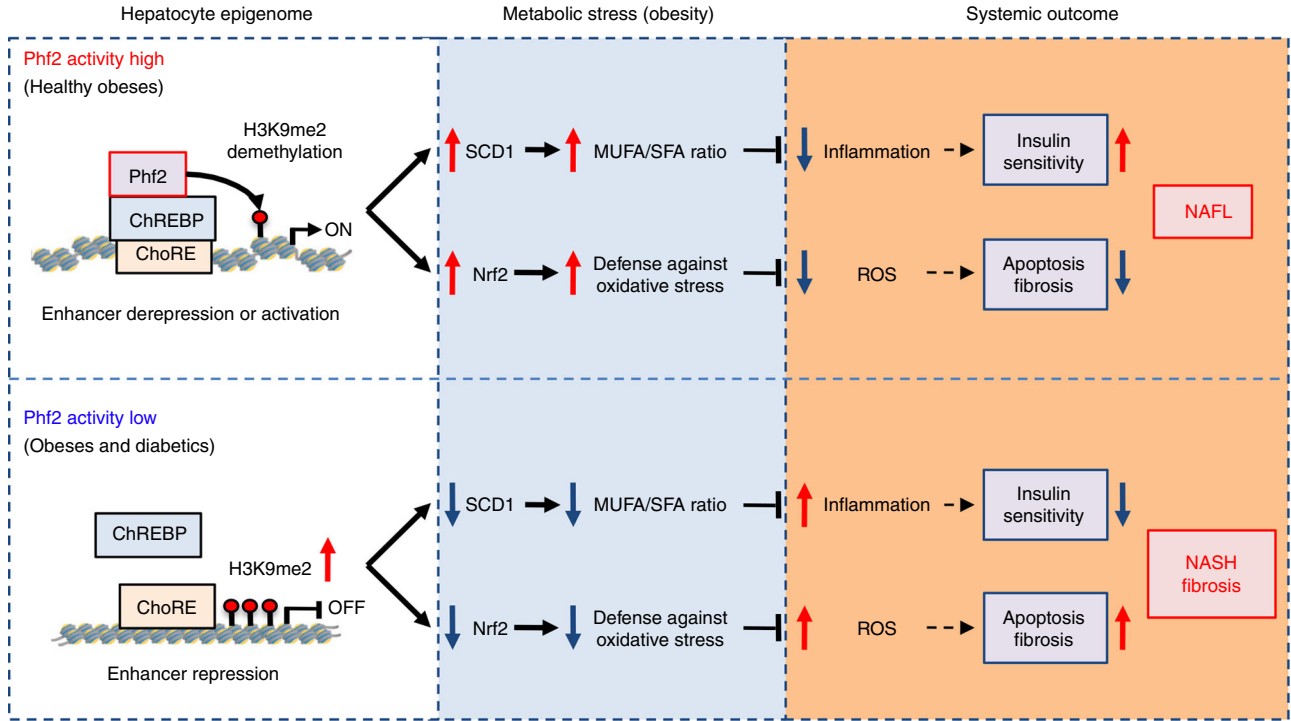

**Fig. 10** Schematic representation of Phf2 action in NAFL development and progression. Phf2 activation, through H3K9me2 demethylation at specific ChREBP-regulated gene promoter, increases MUFA synthesis and anti-oxidative stress defenses. Consequently, Phf2 activation protects liver from inflammation, insulin resistance, oxidative stress, and fibrosis

using KOD Hot Start Master mix (Novagen) and cloned into p3XFLAG-CMV-10 plasmid (Sigma). Phf2 enzymatic (H248A) mutant was generated by using Quik-Change Lightning Site-Directed Mutagenesis Kit (Stratagene). Phf2 and Phf2 shRNA (Phf2i) adenovirus were produced by Genecust INC using the adEasy adenoviral vector system. Adenovirus expressing the GFP (Ad-GFP) and ChREBP shRNA (Ad-ChREBPi) were described previously[31]. Adenovirus expressing a shRNA against Nrf2 and associated adenovirus (AAV/DJ) expressing, under the control of the albumin promoter, either GFP, Phf2, an unspecific shRNA (USi), or a shRNA again Phf2 (Phf2i) were purchased from Vector Violabs inc.

**Reporter assay.** The ChoRE-luc (Carbohydrate response element of the L-pyruvate kinase promoter (ChoRE) luciferase, gift from M. Vasseur), the SCD1-luc (1.5 kb of the mouse SCD1 promoter luciferase, gift from J. M Ntambi) and the ARE-Luc (Nrf2-response element luciferase, from promega) were used to measure Phf2 and ChREBP transcriptional activity both in vitro and in vivo. The RSV β galactosidase (RSV β-gal) plasmid was used as constitutive control reporter.

**Cell culture and transfection.** Mouse hepatocytes were harvested, cultured, and infected with adenoviruses as previously described[57]. Hepatocytes were transiently transfected with the appropriate combination of reporters, expression vectors, and control vectors with Lipofectamine 2000® according to the manufacturer's instructions. Twenty-four hours post-transfection, luciferase assays were performed at room temperature. Experimental data are mean of at least three independent experiments with luciferase activity normalized to β-galactosidase activity, conducted in triplicate. Hepatocytes were also infected with Ad-Phf2, Ad-GFP, Ad-Phf2i, Ad-ChREBPi, or Ad-Nrf2i adenovirus (1pfu/cell). Twenty-four hours post-infection, hepatocytes were incubated with either 5 or 25 mM glucose and 100 nM insulin for 18 h.

**Mouse strains and virus injections.** A total of $5 \times 10^8$ plaque forming units (pfu) of Phf2 or GFP adenovirus (Genecust INC) were delivered to 8 weeks old male C57BL/6J (Janvier, France) or Nrf2 knockout mice (RIKEN BRC) by tail vein injection as previously described[15]. Associated adenoviruses encoding GFP, FLAG-tagged Phf2 (FLAG-Phf2), USi or Phf2i under the control of the albumin promoter were generated by Vector Biolabs inc using the AAVDJ/8 serotype. A total of $5 \times 10^{11}$ vg of either AAVDJ/8-GFP, AAVDJ/8-F-Phf2, AAVDJ/8-USi, or AAVDJ/8-Phf2i were delivered to 8 weeks old male C57BL/6J by tail vein injection. The numbers of mice used in each experiment were selected based on the expected variations between animals and variability in adenovirus or AAV injections. One week after treatment, mice were fed with a normal chow diet (NCD) or with a high fat and high sucrose diet (HFHSD) for 16 weeks. All mice were adapted to their environment for 1 week before study and were housed in colony cages with 12 h

light/dark cycle in a temperature-controlled environment. All procedures were carried out according to the French guidelines for the care and use of experimental animals. All animal studies were approved by the "comité d'éthique pour l'expérimentation animale" of the University of Paris Descartes and by the mouse core facility of the Cochin institute (CEEA34.AFB/CP.082.12). Mice had free access to water and regular diet (NCD) (65% carbohydrate, 11% fat, 24% protein) or high fat and high sucrose diet (HFHSD) (47.6% carbohydrate, 23.2% fat, 17.3% protein) unless otherwise specified. No method of randomization or blinding was used in any of the experiments. Mice that do not overexpressed GFP or Phf2 or in which Phf2 expression was not inhibited after either adenovirus or AAV treatment were removed from the experimental cohort.

**Metabolic tests.** Glucose tolerance tests were performed by oral administration of glucose (1 g D-glucose/kg body weight) after overnight fasting. Insulin tolerance tests were performed by intraperitoneal (ip) injection of human insulin (0.75 unit insulin/kg body wt, Actrapid Penfill, NovoNordisk) 5 h after food removal. Pyruvate tolerance tests were performed by ip injection of sodium salt pyruvate (2 g/kg body weight) after an overnight fast. Blood was collected from the tail vein in a heparinized capillary tube and glucose concentration was determined using the One-Touch AccuChek Glucometer (Roche).

**Analytical procedures.** Blood glucose values were determined using an AccuChek Glucometer (Roche Diagnostic Inc.). Serum triglycerides (TG), free fatty acids (NEFA), alanine aminotransferase (ALAT) and aspartate aminotransferase (ASAT) concentrations were determined using an automated Monarch device (Laboratoire de Biochimie, Faculté de Médecine Bichat, France). Plasma insulin concentrations were determined using a rat insulin ELISA assay kit (Crystal Chem) using a mouse insulin standard. Glycogen concentrations were determined in liver extracts as previously described[15]. Liver TG, diacyglyceride (DAG), and cholesterol ester contents were extracted using the Folch's method with chloroform/methanol (2:1, v/v) and their subsequent separation by thin-layer chromatography (TLC) on silica-gels plates (Merck Chemicals) using petroleum ether/diethyl ether/acetic acid (85:15:0.5, v/v/v) as the mobile phase. Lipids were visualized with iodine vapor. Bands were scraped and TG, DAG, and cholesterol esters were extracted from silica by chloroform/methanol and measured with a colorimetric diagnostic kit (Triglycerides FS; Diasys). Oxidative stress was assessed by measuring levels of protein carbonylation using the OxyBlot Protein Oxidation kit (Millipore). Gpxs activities were determined both in vivo and in vitro using the Glutathione Peroxidase Activity kit according to the manufacturer's instructions (BioVision). The NADP/NADPH ratio was determined using the NADP/NADPH assay kit (Abcam). Reduced (GSH) and oxidized (GSSR) glutathione contents were assessed in liver extracts using the GSH/GSSG-Glo assay kit (Promega). Cytotoxicity, viability, and

caspase activation were measured using the ApoTox-Glo assay kit (Promega). In vitro ROS production was measured using the amplex® red Hydrogen Peroxide/Peroxidase Assay Kit according to the manufacturer's instructions (Life-technologies). Proteasome peptidase activity was determined as previously described[58].

**Metabolite analysis**. GC-MS metabolomic analysis was performed by Matabolon INC from our in vivo study, in which Phf2 or GFP were overexpressed in the liver of C57BL/6J mice for 3 weeks (15 mice per group). Metabolites were extracted by 80% methanol at −20 °C and dried by vacuum centrifugation. GC-MS analysis was performed with a Waters GCT Premier mass spectrometer fitted with an Agilent 6890 gas chromatograph and a Gerstel MPS2 autosampler. Data were collected using MassLynx 4.1 software (Waters). Metabolites were identified and their peak area was recorded using QuanLynx. Data were normalized for extraction efficiency and analytical variation by mean centering the area of D4-succinate.

**Measurement of glucose and oleate metabolic fluxes on liver explants**. Liver explants from fed GFP or Phf2 mice (200 mg) were incubated in duplicate in 25 mL conical glass vial sealed with rubber caps containing plastic center wells in 3 mL of Krebs-Henseleit bicarbonate buffer (pH 7.4) at 37 °C for 2 h. For oleate metabolism, oxidation, and esterification rates were determined after incubation in presence of 0.3 mM [1–14C] oleate (0.5 Ci/mol) bound to 1% (w/v) free fatty acid bovine serum albumin. For glucose metabolism, oxidation, and esterification rates were assayed after incubation with 25 mM [U-14C] D-glucose (μCi/mmol).

**$^{14}CO_2$ production measurement**. At the end of the incubation time, the media is transferred to a new conical glass vial for $CO_2$ production measurement and the liver explants are washed three times in ice cold PBS before processing them for lipid extraction and analysis. Perchloric acid is injected into the incubation media through the rubber cap to a final concentration of 4% (v/v). Benzethonium hydroxide is injected through the rubber cap into a plastic well suspended above the incubation media. During 1 h of vigorous shaking at 25 °C, the released [14C] $CO_2$ is trapped by the benzethonium hydroxide. [14C] $CO_2$ release is then assessed by scintillation counting.

**$^{14}C$ incorporation into intracellular lipids and TAG quantification**. Fatty acid esterification rates were measured in the remaining liver piece or from primary cultured hepatocytes by tracing newly synthesized triglycerides from $^{14}C$-oleate or $^{14}C$-glucose. $^{14}C$-TGs were extracted using chloroform/methanol (2:1, v/v) and separated by TLC as previously described and counted in scintillation liquid.

**Hepatic triglyceride secretion rate**. Mice were fasted for 4 h and ip injected with 400 μl of 7.5% tyloxapol solution (Sigma) in PBS. Blood was collected over a time course and plasma triglyceride content was quantified using a triglyceride assay kit (Triglycerides FS; Diasys).

**Lipidomic analysis**. Fatty acid profiling was performed at the lipidomic core facility of Toulouse (INSERM, Metatoul). Briefly, after homogenization of tissue samples in methanol/5 mM EGTA (2:1 v/v), lipids corresponding to an equivalent of 1 mg of tissue were extracted in chloroform/methanol/water (2.5:2.5:2.1, v/v/v), in the presence of internal standards: 1,3-dimyristine (for DAG) and glyceryl tri-nonadecanoate (for TG). Chloroform phases were evaporated to dryness. Neutral lipids were purified over an SPE column (Macherey Nagel glass Chromabond pure silice, 200 mg): after washing cartridge with 2 mL chloroform, lipid extract was applied on the cartridge in 20 μL chloroform, and neutral lipid were eluted with chloroform/methanol (9:1, v/v; 2 mL). The organic phase was evaporated to dryness and dissolved in 20 μL ethyl acetate. A sample (1 μL) of the lipid extract was analyzed by gas-liquid chromatography on a FOCUS Thermo Electron system, using Zebron-1 (Phenomenex) fused silica capillary columns (5 m × 0.32 mm inside diameter [i.d.], 0.50 μm film thickness). Oven temperature was programmed from 200 to 350 °C at a rate of 5 °C/min, and the carrier gas was hydrogen (7.25 psi). The injector and the detector were at 315 and 345 °C, respectively. To assess SCD1 activity, the Δ9 desaturation index was calculated as the abundance of SCD1 products (palmitoleic and oleic acids) relative to both SCD1 products and substrates (palmitic and stearic acids). Ceramide profiling was performed by metabolon, Inc. (Durham, North Carolina, USA) from 30 mg of frozen liver tissue.

**Isolation of liver mitochondria**. All steps were carried out at 4 °C. Fresh tissues were minced in a sucrose buffer containing 0.3 M sucrose, 5 mMTris/HCl, and 1 mM EGTA (pH 7.4). Minced tissue was carefully disrupted in a Thomas' potter at low speed rotation. Unbroken cells and nuclei were removed by two successive centrifugations of the homogenate at 750 g for 10 min. Mitochondria were collected after centrifugation of the supernatant at 10,000g for 20 min, and protein content was assayed using a bicinchoninic acid kit (Sigma).

**Cellular respiration of liver homogenate**. One hundred μL of homogenate (supernatant after centrifugation at 750 g) were added into 2 mL of sucrose buffer

(100 mM KCl, 40 mM sucrose, 5 mM MgCl2, 1 mM EGTA, BSA 4 mg/mL and 5 mM KPi, pH 7.4) and were introduced in the respiratory chamber of an oxygraph O2 k (Oroboros) at 25 °C. Cellular respiration was determined under glutamate/malate condition (GM 5 mM), with the addition of ADP (50 μM to 2.65 mM), in condition oligomycin (2 μM), and finally with cyanide (1 mM KCN). Cellular respiration was also determined under succinate condition (Succ 7 mM), with the addition of rotenone (1 μM) to inhibit OXPHOS complex I, ADP (50 μM–2.65 mM), in condition oligomycin (2 μM), and/or increasing amounts (2.5–12 μM) of carbonyl cyanide m-chlorophenylhydrazone (CCCP) and finally with cyanide (1 mM KCN). For both experiments using either glutamate/malate or succinate, mitochondrial respiration was determined with substrate and ADP. Leak was calculated in the presence of oligomycin that inhibits ATP synthase.

**Measurement of PKCε activity**. PKCε was immunoprecipitated from liver samples of GFP or Phf2 overexpressing mice using 0.5 μg of polyclonal rabbit anti PKCε antibody (Santa Cruz, CA). After overnight incubation at 4 °C, Gamma Bind Sepharose was added and rotated for 2 h, and the immunocomplex containing PKCε was pelleted by brief centrifugation. After three washes, the pellets were resuspended in kinase buffer (50 mM HEPES, 100 mM NaCl, 10 mM MgCl2, 50 mM NaF, 1 mM NaVO4, 1 mM dithiothreitol, and 0.1% Tween 20). The in vitro kinase reaction was initiated by addition of 40 μg/ml phosphatidylserine/reaction, 10 mM MgCl2, 0.25 mM ATP (cold) and 1 μCi [γ-32P]ATP (10 mCi/mmol), and 10 μg of recombinant histone IIIS as a substrate. After 20 min of incubation at 37 °C, the kinase reactions were terminated by adding SDS-PAGE sample buffer and heating at 95 °C for 5 min. Phosphorylated histone IIIS protein was then separated on 12% SDS-PAGE and transferred to Hybond nitrocellulose membranes. The bands corresponding to phosphorylated histone IIIS were detected by auto-radiography. As positive control of our PKCε activity measurement, PKCε was also immunoprecipitated from liver samples of C57Bl6/J mice fed with either standard (NCD) or HFHS diet for 18 weeks. HFHS diet has been previously shown to induce PKCε activity in the liver[59].

**DAG measurement at the plasma membrane and in the cytosol**. Hepatic DAG content was separated into membrane and cytoplasmic fractions as previously described[60]. Briefly, liver samples (250 mg) were homogenized on ice using a dounce homogenizer in lysis buffer containing 10 mM Tris base, 0.5 mM EDTA, 250 mM sucrose, and protease inhibitors. Then 3% sucrose was layered on top of the homogenate, and samples were centrifuged at 100,000g for 1 h at 4 °C. The supernatant and lipid layers were removed and designated as the cytoplasmic fraction. The pellet, designated as the membrane fraction, was resuspended in homogenization buffer for DAG analysis. Concentrations of DAG in each fraction were determined by using mass spectrometry analysis as previously described.

**Staining techniques**. For histology studies, tissues were fixed in 10% neutral buffered formalin and embedded in paraffin. Then, 5 μm sections were cut, rehydrated through descending grades of ethanol, and stained with hematoxylin eosin (HES) or were subjected to immunohistochemical analysis using antibodies against F4/80, type I collagen, and α-SMA. To evaluate liver fibrosis, sections were stained with Masson's trichrome (Sigma Aldrich) according to the manufacturer's instructions. The degree of hepatic fibrosis was quantified by morphometric analysis. In addition, apoptotic bodies (councilman bodies) were counted in ten consecutive fields from HES slides. For the detection of neutral lipids, liver cryo-sections were stained with the Oil Red O technique, using 0.23% of Oil Red O dye dissolved in 65% isopropyl alcohol for 10 min as previously described[15].

**In vivo bioluminescence imaging**. For measurement of Phf2 and ChREBP transcriptional activity in vivo, the ChoRE-luc, SCD1-luc, and ARE-luc plasmids were overexpressed with the RSV β galactosidase construct through hydrodymanic gene transfer in the liver of C57BL/6J mice (10 μg each) as previously descried[61]. Mice were studied in fed state 48 h after injection. For imaging, mice were ip injected with 100 mg/kg of sterile firefly D-luciferin (Caliper). After 10 min, mice were ip injected with a mix of ketamine/xylazine, imaged on the PhotonImager system (Biospace Lab) and analyzed with the M3Vision software (Biospace Lab). For in vivo measurement of ROS ($H_2O_2$ levels), using the PCL-2 probe, mice were infected with both adenovirus overexpressing the luciferase and the RSV β-gal (5 × $10^8$ pfu each). Mice were studied 1 week later in fed state. The PCL-2 probe was used as previously descried[62]. Briefly, mice, overexpressing the luciferase in the liver, were anesthetized with isoflurane and ip injected with a mixture of the PCL-2 and D-cysteine (0.05 μmol each, in 150 μL of 1:1 DMSO:PBS). Following injections, mice were imaged with the PhotonImager system (Biospace Lab). The following setting was used, 1 min exposure, 60 images (60 min total). Imaging is started immediately following injection of the probe. For all in vivo imaging experiments, the RSV β-gal plasmid was used as a constitutive control reporter to normalize the bioluminescence signal.

**β-galactosidase determination**. β-gal assays for normalization of the ChoRE, SCD1, ARE-luc activity, and PCL2 probe were performed using 10 μL of hepatocyte lysates, 50 μL 2× buffer (1.33 mg/ml 2-Nitrophenyl β-D-galactopyranoside, 100 mM 2-mercaptoethanol, 2 mM Magnesium Chloride, 200 mM sodium

phosphate pH 7.5, all purchased from Sigma, St. Louis MO), and 40 µL water in each well of a clear 96-well plate. The plate was covered and incubated 30 min at 37 °C, and absorbance at 405 nm was determined with a xenius plate reader (Safas). Lysate samples were assayed in triplicate. Lysates from unstransfected cells were used as controls for background activity. β-Gal activity was expressed as A405 units/mg protein.

**Liver lysates**. Livers were frozen in liquid nitrogen and kept at –80 °C until use. Mouse tissues were sonicated three times for 10 s each at 4 °C in lysis buffer [150 mM NaCl, 50 mM Tris-HCl pH 7.5, 5 mM EDTA, 30 mM Sodium pyrophosphate, 30 mM Sodium Fluoride, 1% Triton × 100, and protease inhibitor cocktail (Sigma, St. Louis, MO)]. Lysates were centrifuged 16,000g at 4 °C for 20 min and supernatants were reserved for protein determination, and SDS PAGE analysis.

**Preparation of nuclear fractions**. Hepatocytes were wash two times with ice cold PBS. Cells were collected in PBS and pelleted at 3000 rpm at 4 °C. Cell pellets were resuspended in hypotonic lysis buffer (50 mM Hepes ph 7.4, 10 mM NaF, 1 mM EDTA, 0.5 mM DTT with protease inhibitors). Cells were lysed using a dounce homogenizer and centrifuged at 4200 rpm for 10 min at 4 °C. Cytosolic supernatants were collected. Nuclear pellets were washed three times in 1 mL of hypotonic lysis buffer and resuspended in 200 µL of nuclear extraction buffer (50 mM Hepes pH 7.4, 420 mM NaCl, 10 mM NaF, 1 mM EDTA, 0.5 mM DTT). Samples were sonicated and centrifuged at 13,000 rpm for 30 min. Nuclear supernatants were collected.

**ChIP-sequencing**. All the ChIP-Seq experiments were performed using samples collected from primary hepatocytes. ChIP-Sequencing (ChIP-seq) sample preparation and computational analysis of Illumina GA I/II data were performed by Actif Motif INC as follows. Primary cultured hepatocytes were incubated 24 h in the presence of 25 mM glucose and 100 nM insulin and were subjected to standard ChIP, using indicated antibodies, as described above. ChIP DNA-samples were then subjected to preparation for ChIP-Seq library construction: the libraries were constructed following Illumina's Chip-Seq Sample prep kit. Briefly, Chip DNA was end-blunted and added with an "A" base so the adapters from Illumina with a "T" can ligate on the ends. Then 200–400 bp fragments are gel-isolated and purified. The library was amplified by 18 cycles of PCR. Primary analysis of ChIP-Seq data sets: the image analysis and base calling were performed by using Illumina's Genome Analysis pipeline. The sequencing reads were aligned to the mouse genome UCSC build by using BOWTIE32 alignment programs in two ways: only uniquely aligned reads were kept or both uniquely aligned reads and the sequencing reads that align to repetitive regions were kept for downstream analysis (if a read aligns to multiple genome locations, only one location is arbitrarily chosen). The multiple reads were collapsed in order to reduce the PCR biases. The aligned reads were used for peak/island finding with MACS33. MACS peak/island predictions were adjusted for genome instabilities (amplifications, deletions) either by considering a local background area (MACS) that was used as a reference for the subsequent calculation of the enrichment scores. Annotating and comparing the ChIP-Seq peaks: the ChIP-Seq peaks were mapped on the UCSC genome browser. A peak was considered to be associated with a particular genome feature (for example, promoter, intron, and exon) if the peak summit (MACS peaks) was located within 3 Kb distance of TSS, or within an exon or intron. If a peak intersected with multiple genome features, all the corresponding genome features were considered when computing the genome distributions. Phf2 peaks were considered common if the predicted peaks intersected over at least 1 bp. The gene ontology analysis was carried out by using DAVID/EASE35 and the sequence motif enrichment analysis was performed by using HOMER. For motif finding, we used MEME Suite with the default settings except that the expected motif site is any repetitions and the find uncentered regions option is selected. Logos of different motifs were generated from MEME-ChIP analysis. We used Regulatory Sequence Analysis Tools to generate the final consensus sequence logo.

**Chromatin immunoprecipitation**. For ChIP-seq validation, chromatin immunoprecipitation (ChIP) was performed as described[63]. Briefly, primary cultured hepatocytes or liver samples were cross-linked with 1% formaldehyde (Sigma) and chromatin DNA was sheared to 300–500 bp average in size through sonication. Resultant samples were immunoprecipitated with control IgG or specific antibodies overnight at 4 °C and followed by incubation with protein A/G magnetic beads (Ademtech) for an additional 2 h. After washing and elution, the protein–DNA complex was reversed by heating at 65 °C overnight. Immunoprecipitated DNA was purified by using QIAquick spin columns (Qiagen) and analyzed by qPCR using a Roche Light Cycler. Primers used are specific for regions tested and their sequences are available on request. All ChIP were repeated at least three times and representative results were shown. All signals were normalized to input chromatin signals. Chromatin accessibility was assessed using the EpiQ chromatin analysis kit according to the manufacturer's instructions (Bio Rad).

**RNA profiling**. Phf2 or GFP were overexpressed through adenoviral gene delivery in the liver of C57BL/6 J mice. After 3 weeks, total RNA was isolated from liver using RNeasy Mini Kit (Qiagen) following the manufacturer's protocol. The

microarray experiments and data normalization were performed by the Cochin Institute transcriptomic core facility. Briefly, RNA profiling was performed using Affymetrix GeneChip Human Gene 2.0 ST array, which interrogates 25,000 gene sequences. Raw data were normalized using the Robust Multichip Algorithm (RMA) in Bioconductor R software. Then, all quality controls and statistics were performed using Partek GS software. First, hierarchical clustering (Pearson's dissimilarity and average linkage) and principal composant analysis (PCA) were performed as unsupervised exploratory data analysis. Then, a classical analysis of variance (ANOVA) for each gene and pair wise Tukey's post-hoc tests between groups were conducted to find differentially expressed genes. Finally, p-values and fold changes were used to filter and select differentially expressed genes. Interactions, pathways and functional enrichment analysis were carried out through the use of IPA (Ingenuity Systems, USA www.ingenuity.com) and DAVID/EASE tools (http://david.abcc.ncifcrf.gov/).

**Isolation of total RNA and analysis of mRNA expression by quantitative PCR**. Total cellular RNAs from whole liver or from primary cultured hepatocytes were extracted using the RNeasy kit (Qiagen). mRNA levels were then measured as previously described using a Roche Light Cycler[64]. Primer sequences are available on request.

**Western blotting and lysine demethylation assays**. Western blot were carried out as previously described[65]. Revelation and quantification were performed using the Chemidoc MP system instrument (Bio-Rad). For preparation of Phf2 IP from mammalian cells, 293T cells were transfected with Flag-tagged Phf2 plasmid with Lipofectamine 2000® (Invitrogen). After 24 h, the cells were lysed with TNE lysis buffer (20 mM Tris at pH 7.8, 450 mM NaCl, 0.5 mM EDTA, 1 % NP40). For demethylase reaction, purified Flag-Phf2 protein was mixed with different substrates (recombinant histone H3 (2 µg) (Upstate), synthetic biotinylated histone H3 peptides either mono, di, or trimethylated at the K9 position (ARTKQ-TARKSTGGKAPRKQLATKAARKSAPATGGVKKPHR-YC-Ttds-K-Biotin, JPT, Germany), purified mononucleosome (10 µg) or core histone) in the reaction buffer (final volume 20 µL) (20 mM Tris-HCl at pH7.5, 150 mM KCl, 50 µM Fe (NH4)2(SO4)2–6H2O, 1 mM α-ketoglutarate, 1 mM ascorbate, 20 µM ZnCl2). The mixtures were incubated at 37 °C for 12 h, terminated by boiling for 5 min in SDS sample buffer at pH 7. Histone modifications were detected by specific antibodies, as described above.

**Histone peptide pull-down assay**. Biotinylated histone peptides (Millipore and Abgent), either modified or unmodified, were pre-incubated with Dynabeads M-280 streptavidin blocked with BSA (Invitrogen) at room temperature for 1 h and then washed with washing buffer (0.01% Tween-20 in PBS) twice before mixing with cell exacts expressed proteins (FLAG-tagged WT Phf2 or Phf2ΔPHD). After overnight incubation at 4 °C in 20 mM HEPES pH8.0, 1.5 mM MgCl2, 0.15 M NaCl, 25% glycerol, 1 mM DTT, 0.2 mM EDTA, 0.01% Tween-20, pull-downs were washed with washing buffer (same as incubation buffer) five times before resolving by 4–12% Bis-Tris gel (Invitrogen) and analysis by immunoblotting.

**Immunofluorescence**. Primary hepatocytes, transfected with expression vectors as indicated, were directly fixed for 5 min with 4% formaldehyde (v/v) in Phem buffer (60 mM PIPES, 25 mM HEPES, 10 mM EGTA, 2 mM MgCl2, pH6.9) and then permeabilized for 2 min with 0.1% Triton X-100 in Phem buffer. After three washes with Phem buffer, blocking solution (1% BSA in PBS, pH 7.4) was applied for 30 min and primary antibodies against Flag and H3K9me2 were added in blocking buffer for 1 h at room temperature. After three washes with PBS/0.1% Triton X-100, cells were incubated with DAPI and with secondary antibodies conjugated with fluorescent dyes for 1 h, washed again with PBS/0.1% Triton X-100, and mounted in 20 mM Tris, pH 9/0.1% p-phenylenediamine. Image intensities for each antibody were scaled identically.

**Human liver biopsies**. We analyzed liver samples from white morbidly obese French patients who underwent abdominal surgery and were included in the Biological Atlas of Severe Obesity (ABOS) cohort realized at the "Centre Hospitalier Régional Universitaire de Lille", France (ClinicalGov NCT01129297)[66]. Informed consent was obtained from all individuals and the experimental design was approved by the Hospital's Ethics Committee. All patients underwent preoperative evaluation (medical history and physical examination) and were devoid of any condition that could have increased abdominal surgery risks. For this study, to minimize potential confounding factors, only men individuals of similar age range were selected for this study. Three groups were defined and staged as lean controls (no steatosis), obese subjects with liver steatosis but limited inflammation (NAS ≤ 3) and obese subjects with NASH (NAS ≥ 5). The degree of steatosis was defined as the percentage of hepatocytes containing fat droplets. For this study, diagnosis was made for samples exhibiting >50% of fatty acid infiltration within the hepatocytes in both steatotic and NASH liver groups (macrovesicular steatosis). Histological features was further blindly evaluated by two pathologists (P.M and F. P) using the NAFLD activity score (NAS), as recommended by the NASH clinical network[67] (see Table S4 and S5 for scoring interpretation). The NAS is defined as the unweighted summary of scores for percentage of steatosis (0: 0–5%; 1: 5–30%;

2: 30–66%; 3: 66–90%), hepatocyte ballooning (0: none; 1: few balloon cells; 2: many cells/prominent ballooning) and lobular inflammation (0: no foci; 1: <2 foci/200×; 2: 2–4 foci/200×; 3: >4 foci/200×) thus ranging from 0 to 8. A NAS score of 5 or more was correlated with NASH and a NAS score of less than 3 was considered as the absence of NASH[67]. Fibrosis stages were also determined separately from NAS score (F0: none; F1: perisinusoidal or periportal; F1a: mild, zone 3, perisinusoidal; F1b: moderate, zone 3, perisinusoidal; F1c: portal/periportal; F2: perisinusoidal and portal/periportal; F3: bridging fibrosis; F4: cirrhosis) (supplementary Table 4). Type 2 diabetic patients were defined by fasting plasma glucose levels ≥07.0 mmol/L; normoglycemic individuals were defined by fasting glucose levels of <5.6 mmol/L. HOmeostasis Model Assessment of Insulin Resistance (HOMA-IR) was assessed with the homeostasis model assessment and calculated according to Wallace et al[68]. Needle liver biopsies were performed in the left hepatic lobe through laparoscopy, within 15 min from the beginning of the procedure. Samples were immediately frozen in liquid nitrogen and stored at −80 °C.

**Statistical analyses**. Results are expressed as mean ± SEM, and were analyzed with analysis of variance using GraphPad Prism software. Sample sizes (n) were reported in the corresponding figure legend. All experiments were performed on at least three independent occasions. No statistical method was used to predetermine sample size. After the normal distribution was confirmed with the Kolmogorov–Smirnov test, statistical comparisons between two groups were performed using a student's t test followed by Mann-Whitney post hoc test. Comparisons among multiple parameters were performed by two-way ANOVA followed by Bonferroni's post hoc comparisons. We did not estimate variations in the data. The variances are similar between the groups that are being statistically compared. In all cases, P values less than 0.05 were considered significant.

**Data availability**. The authors declare that all data supporting the findings of this study are available within the paper and its supplementary information files, or are available from the corresponding author upon reasonable request. Microarray analysis can be found at the Gene Expression Omnibus database under accession number GSE61575 and GSE61576.

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

## Acknowledgements

The authors would like to thank Dr Anne-Françoise Burnol, Dr Sandra Guilmeau, and Dr Françoise Levavasseur (Institut Cochin, Département d'Endocrinologie, Métabolisme et Diabète, Université Paris Descartes, Paris, France) for helpful discussions and critical reading of the manuscript. The authors also thank Catherine Esnous for technical assistance, Véronique Fauveau for performing in vivo adenoviral injections, Gilles Renault and the small animal imagery core facility (Institut Cochin, Inserm U1016, Paris) for in vivo bioluminescence imaging, Sébastien Jacques and Florent Dumont from the genomic core facility (Institut Cochin, Inserm U1016, Paris) for microarray bioinformatic analysis, the HistIM core facility (Institut Cochin, Paris, France) for performing liver sections and immuno-staining and Lorenne Robert for managing the Nrf2-KO mice colony. Mice used in this study were housed in an animal facility equipped with the help of the Région Ile de France. The work was performed within the Département Hospitalo-universitaire (DHU) AUToimmune and HORmonal diseaseS (AUTOHORS) and within the Recherche Hospitalo-Universitaire (RHU) "QUID NASH." This study was also supported by grants from the agence nationale de la recherche (ANR-09-JCJC-0057-01, R.D), the French "fondation pour la recherche médicale" (FRM, labélisation équipe, R.D and C.P), "La ville de Paris" (projets emergences, R. D), by the European Commission (ERC starting grant, 336629, LIPIDOLIV; R.D), and by the NIH (GM79465, C.J.C).

## Author contributions

J.B. and R.D. conceived of the hypothesis and designed the experiments. J.B., M.C.A.G., P.E., C.P.B., J.B.M., H.G., and R.D. performed the experiments. J.B., M.C.A.G., P.E., C.P.B., H.G., C.P., and R.D. analyzed and interpreted the data. M.V.W. and C.J.C. produced and provided PCL-2. probes for in vivo $H_2O_2$ measurements. F.C.H. provided Nrf2-KO mice. F.P., V.R. and P.M. provided and characterized human liver biopsies from the ABOS cohort. J.B., J.G., and R.D. wrote the manuscript.

## Additional information

**Competing interests:** The authors declare no competing interests.

