## [Peer Review File · Nature Communications]

Reviewers' comments:

Reviewer #1 (Remarks to the Author):

This study shows that the Plant Homeodomain Finger 2 (Phf2) is a transcriptional co-activator of ChREBP by histone demethylation of the promoter of ChREBP-regulated genes. Phf2 increases synthesis of mono-unsaturated fatty acids with hepatic fat deposition but without inflammation and insulin resistance. Activation of Nrf2 shifts glucose fluxes towards the pentose phosphate pathway and glutathione biosynthesis, and protects against nutrient induced oxidative stress and fibrogenesis. The authors conclude that Phf2 protects from NAFLD progression.

The authors need to be congratulated for the extensive work performed.

Major points

1. Some conclusions are based on the assumption that insulin sensitivity was/remained improved despite elevated liver TGs. Insulin sensitivity derived from glucose/pyruvate tolerance tests was indeed enhanced but no gold standard techniques to simultaneously assess peripheral and hepatic insulin sensitivity were applied. Of note, the higher insulin sensitivity was observed in the presence of markedly lower serum triglycerides. So to what extent was the observed improvement simply due to differences in peripheral TG and FFA.
2. Furthermore, the conclusion (see last figure) on the prevention from insulin resistance in this model remains unclear. Higher DAGs should result in stimulation of PKCepsilon activity, which did not seem to be the case. So was DAG present in different subcellular compartments and did you really measure activity or simply translocation of PKCepsilon?
3. How should MUFA prevent from insulin resistance according to the model described by the authors.
4. The clinical relevance of the proposed mechanisms remains illusive, because nrf2 mRNA expression and antioxidant activity were not increased in NASH patients.

Minor points

1. The Introduction should describe more Phf2 may be important for NAFLD and to what extent this may be of clinical relevance in humans, rather than summarizing the results for the paper in the second paragraph.
2. Figure 8G: NAFLD comprises all liver diseases, so the authors obviously wanted to state NAFL or simple steatosis.

Reviewer #2 (Remarks to the Author):

Bricambert et al. report that the histone demethylase Phf2 contributes to certain metabolic dysregulation associated with non alcoholic fatty liver disease. The authors identified Phf2 as a potential coactivator of the transcription factor ChREBP, a known fed-induced activator of the lipogenic pathway. Through a series of experiments in-vitro and in-vivo the present studies provide sufficient results that support, at least in part, the conclusions raised in the manuscript. To strengthen the manuscript the authors need to address the following points:

1. The authors have described a Phf2 H248A mutant which has impaired activity; however they have not used this in majority of the experiments. Use of this mutant in most of the experiments,

whether in vivo (for example Fig 7), or those in primary hepatocytes (for example Fig 4) would have been a good negative control that would strengthen the authors' conclusions.

2. The authors have shown that liver-specific Phf2 overexpression protects mice from diet-induced obesity, insulin resistance and fibrinogenesis. What are the systemic effects of this overexpression? Are the muscle and adipose tissues also sensitized to insulin? Or are the observed effects solely due to the liver?

3. Phf2 expression also appears to be significant in the pancreas, adipose tissue and muscle – have the authors assessed if a similar regulation of occurs in these tissues either in vitro or in vivo, as this would be informative.

4. Is Phf2 also known to methylate proteins? If so, do the authors think that some of this signaling could also feed into the observed effects? A discussion on this would be useful.

5. In Figure 1E, are there certain histone marks that do not change? If there are several being altered (such as H3K4me3), it is unclear if the transcription of the gene corresponds solely to H3K9me2 demethylation.

6. Are the same effects of Phf2 overexpression in vivo in Fig 3 also observed in primary hepatocytes?

Reviewer #3 (Remarks to the Author):

In this manuscript, the authors demonstrate the metabolic reprogramming through Plant homeodomain finger 2 (Phf2). They identified Phf2, a histone demethylase for H3K9me2, as a co-activator of carbohydrate-responsive element binding protein (ChREBP). Ectopic expression of Phf2 in mouse livers induces gene expression of proteins encoding free fatty acid metabolism, and it also directs glucose into pentose-phosphate pathway and glutathione synthesis through activation of NF-E2-related factor 2. As a result, mice expressing Phf2 in their livers exhibit steatosis associated with a hyper insulin sensitivity state without inflammation. Interestingly, NAFLD progression by high fat and high sucrose diet was greatly suppressed by Phf2-overexpression, which is an Nrf2-activation dependent manner. Further, the authors confirmed the increased level of Phf2 in the livers of obese patients with benign hepatic steatosis. Taken together, the authors concluded that Phf2 protects liver from NAFLD-progression. Below I list a few specific points that should help improve the manuscript by making the dataset more consistent with the claims made by the authors.

Specific comments

1. The authors should perform a tracer study with [¹³C₆] glucose in GFP- or Phf2-expressing wild-type and Nrf2-deficient hepatocytes and present the amount of each labelled metabolite.

2. The level of Nrf2 protein is mainly regulated by post-translational modification (i.e., Keap1-mediated ubiquitination of Nrf2 and subsequent degradation by the 26S proteasome).

Nevertheless, the authors claim simple up-regulation of Nrf2 through Phf2 leads to its activation. The authors should investigate whether ectopic expression of Phf2 has an effect on the Nrf2-activation at the post-translational level or not.

3. Does gene silencing or targeting of Phf2 in mouse livers promote NAFLD-progression? This reviewer thinks that the authors can conduct such experiments using adenovirus system or hydrodynamic gene transfer.

4. The molecular mechanism by which Phf2 activates PI3K/Akt pathway remains unclear.

5. In human obese patients, the level of Phf2 protein fluctuated on the disease-state. Do the authors have any idea to address this?

Minor points

1. Page 7, 11 line from the bottom. Replace "Figure 2D" by "Figure 2C".

2. Page 8, line 13. "Figure 3I" should be "Figure 2I".

3. Figure legend. Figure 1, line 5. Typo: substitute "transected" with "transfected"

4. Figure legend. Figure 6, title. Substitute "fribrogenesis" with "fibrogenesis".

5. Bar graphs shown in Fig. 4C, E and Fig. 5G are difficult to identify. Change the color of bar.
6. In Fig. 5G, is a right graph showing ARE-luc activity correct?

Reviewers' comments:

Reviewer #1 (Remarks to the Author):

This study shows that the Plant Homeodomain Finger 2 (Phf2) is a transcriptional co-activator of ChREBP by histone demethylation of the promoter of ChREBP-regulated genes. Phf2 increases synthesis of mono-unsaturated fatty acids with hepatic fat deposition but without inflammation and insulin resistance. Activation of Nrf2 shifts glucose fluxes towards the pentose phosphate pathway and glutathione biosynthesis, and protects against nutrient induced oxidative stress and fibrogenesis. The authors conclude that Phf2 protects from NAFLD progression.

The authors need to be congratulated for the extensive work performed.

We would like to thank Reviewer #1 for her/his positive and constructive comments. We have now perform additional experiments and answered the points raised.

Major points

1. Some conclusions are based on the assumption that insulin sensitivity was/remained improved despite elevated liver TGs. Insulin sensitivity derived from glucose/pyruvate tolerance tests was indeed enhanced but no gold standard techniques to simultaneously assess peripheral and hepatic insulin sensitivity were applied. Of note, the higher insulin sensitivity was observed in the presence of markedly lower serum triglycerides. So to what extent was the observed improvement simply due to differences in peripheral TG and FFA.

It was not possible, within the frame of this revision period, to perform clamps studies to globally assess peripheral and hepatic insulin sensitivity, since this technique is unfortunately not set up in our Institute. Nevertheless, to evaluate peripheral insulin sensitivity, we collected the epididymal white adipose tissue (WAT) and skeletal muscle of fed liver-specific Phf2 overexpressing mice and conducted western blots analysis to characterize, in those tissues, the activity of the PI3K/Akt signaling pathway. These data are now included in the supplementary Figure 3G of the revised manuscript. These data demonstrate that there is no change in peripheral insulin sensitivity (WAT or skeletal muscle) after liver-specific Phf2 overexpression as compared to control mice, even in the setting of increased circulating TG levels as compared to GFP mice (supplementary Table 1). Overall, these results demonstrate that the overall improved glucose tolerance and insulin sensitivity observed in Phf2 overexpressing mice is primarily the result of enhanced liver insulin sensitivity.

2. Furthermore, the conclusion (see last figure) on the prevention from insulin resistance in this model remains unclear. Higher DAGs should result in stimulation of PKCepsilon activity, which did not seem to be the case. So was DAG present in different subcellular compartments and did you really measure activity or simply translocation of PKCepsilon?

We would like to thank the Reviewer for this question.

In the revised manuscript, we now have isolated and quantified DAG from the plasma membrane and from the cytosol in the liver of GFP and Phf2 overexpressing mice by performing sub-cellular fractionation studies (Figure. 2C). This demonstrates that the DAG content was increased either at the plasma membrane or in the cytosol in the liver of Phf2 overexpressing mice compared to GFP. However, as previously mentioned, no pro-inflammatory response is observed. In the revised manuscript, we also directly measured the PKCepsilon activity (Figure. 2B) in the liver of Phf2 overexpressing mice as a complementary approach to its cellular localization (Supplementary Figure. 3B). Again no difference of PKCepsilon activity was observed between GFP and Phf2 overexpressing mice, clearly dissociating DAG content at the plasma membrane to PKCepsilon activation.

Furthermore, to better explain this apparent dissociation between DAG content at the plasma membrane and the absence of PKCepsilon activation and inflammation, we performed extensive lipidomic analysis on DAG, TG and cholesterol esters species (Figure. 2H). These new data clearly revealed important changes in the Acyl-CoA composition in DAG, TG and cholesterol esters between GFP and Phf2 overexpressing mice. Overall, Phf2 overexpression significantly reduced the concentration of SFA Acyl-CoA content in DAG, TG and cholesterol esters. In the opposite, Phf2 overexpression significantly increased the content of MUFA Acyl-CoA in those lipid species (Figure. 2H). This decrease in SFA content in DAG, TG and cholesterol esters could be instrumental in the protective effects of Phf2 from inflammation and insulin resistance, as lipotoxicity is generally attributed to SFA.

Of note, in the first version of our manuscript (initial Figure. S3E), we globally analyzed the Acyl-CoA composition regardless of the fatty acids species (i.e. DAG, TG or cholesterol esters). In this revised version of our manuscript, we decided to remove this panel, since we believe that the lipidomic analysis performed on DAG, TG and cholesterol esters brings more details as compared to this initial lipidomic analysis. However, if the reviewers or the editor think that this panel is important to our study, we are willing to include this panel again in the revised manuscript.

3. How should MUFA prevent form insulin resistance according to the model described by the authors.

As lipotoxicity is generally attributed to SFA, their conversion into specific MUFA, in response to Phf2 overexpression, could be directly instrumental in the protective effect of Phf2 from inflammation and insulin resistance, by reducing the intracellular concentration of SFA. To demonstrate the consequences of Phf2-mediated SFA desaturation in these effects, we now have included in our revised manuscript a lipidomic analysis performed on primary cultured hepatocytes overexpressing Phf2 and in which SCD1 expression was inhibited. We also characterized more precisely the pro-inflammatory signaling pathway by western blots (Figure. 2I and J).

In control hepatocytes, palmitate treatment increases the intracellular content of SFA,

which is correlated with the induction of pro-inflammatory signaling pathway and development of insulin resistance, as evidence by increased JNK, IKKb or NFkB phosphorylation and decreased AKT or ERK phosphorylation (Figure. 2I, J and K). However, Phf2 overexpression, by increasing SFA desaturation, decreases SFA content in hepatocytes after palmitate treatment (Figure. 2J). In this context, Phf2 overexpression prevents palmitate-induced inflammation and insulin resistance (Figure. 2I and K). In contrast, in a context of SCD1 deficiency, Phf2 overexpression is no longer able to decrease SFA content in hepatocytes by converting them into MUFA (Figure. 2J). Consequently, SFA content is significantly increased under palmitate treatment. In this context, inflammation and insulin resistance are enhanced even in the setting of Phf2 overexpression (Figure. 2I and K).

Overall, our results demonstrate that this is not MUFAs per se that would prevent inflammation and insulin resistance but most likely the decrease in SFAs content after their specific desaturation by the SCD1 enzyme.

4. The clinical relevance of the proposed mechanisms remains illusive, because nrf2 mRNA expression and antioxidant activity were not increased in NASH patients.

In the revised version of our manuscript, we specifically inhibited Phf2 expression in the liver of C57Bl6/J mice through AAV strategy. We then challenged these mice on HFHS diet (HFHSD) for 12 weeks (Figure. 7). Mice with targeted disruption of Phf2 expression in the liver (Phf2i) are more prone to develop fibrosis on HFHSD compared to control mice (USi). These results further demonstrate that reduced Phf2 activity in the liver favors the progression of NAFLD into fibrosis during the physiopathology of obesity (Figure. 7C and D). These effects are directly associated with reduced nrf2 activity as revealed by profound defects in stimulating anti-oxidative stress response under Phf2 deficiency (Figure. 7G and H). Overall, we think that reduced Phf2 activity in both mice and NASH patients favors fibrosis development by decreasing nrf2 expression and activity and as a consequence anti-oxidative stress response. Overall, pharmacological activators of Phf2, through facilitating either lipid partitioning and stimulating anti-oxidative stress defenses, could be an interesting and promising pharmacological approach to protect liver from the pathogenesis accumulation of lipids during NAFLD development and progression. In this line of evidence, the fact that liver-specific Phf2 overexpression can counteract the effects of HFHSD feeding on liver fibrogenesis is a proof of concept that stimulating Phf2 H3K9me2 histone demethylase activity could be important to promote nrf2-mediate anti-oxidative stress defenses (Figure. 8).

Minor points

1. The Introduction should describe more Phf2 may be important for NAFLD and to what extent this may be of clinical relevance in humans, rather than summarizing the results for the paper in the second paragraph.

We have changed the introduction of our manuscript accordingly by introducing the role of Phf2 in the control of lipid metabolism in adipose tissue and to what extent Phf2 may be important for NAFLD development and/or progression in this context.

2. Figure 8G: NAFLD comprises all liver diseases, so the authors obviously wanted to state NAFL or simple steatosis.

We thank the reviewer to point this inaccuracy. We have corrected this imprecision in the revised version of our manuscript (Figure. 9G).

Reviewer #2 (Remarks to the Author):

We would like to thank Reviewer #2 for her/his positive and constructive comments. We have now perform additional experiments and answered the points raised.

Bricambert et al. report that the histone demethylase Phf2 contributes to certain metabolic dysregulation associated with non alcoholic fatty liver disease. The authors identified Phf2 as a potential coactivator of the transcription factor Chrebp, a know fed-induced activator of the lipogenic pathway. Through a series of experiments in-vitro an in-vivo the present studies provide sufficient results that support, at least in part, the conclusions raised in the manuscript. To strength the manuscript the authors need to address the following points:

1. The authors have described a Phf2 H248A mutant which has impaired activity; however they have not used this in majority of the experiments. Use of this mutant in most of the experiments, whether in vivo (for example Fig 7), or those in primary hepatocytes (for example Fig 4) would have been a good negative control that would strengthen the authors' conclusions.

We thank the reviewer for this useful suggestion. In our revised manuscript, we now have overexpressed, in primary cultured hepatocytes, either the wt form of Phf2 or its H248A mutant to determine the importance of Phf2 histone demethylase activity in regulating ChREBP transcriptional activity and anti-oxidative stress response. These results are now included in the supplementary figures 6 and 7. Compared to Phf2 wt, our results demonstrated that Phf2 H248A overexpression, which lacks its histone demethylase activity, was unable to either stimulate ChREBP or nrf2 transcriptional activity to enhance glycolytic, lipogenic and anti-oxidative stress program (supplementary figures 6 and 7). Overall, these results clearly demonstrate that Phf2 histone demethylase activity is essential to mediate its protective effect in hepatocytes.

2. The authors have shown that liver-specific Phf2 overexpression protects mice from diet-induced obesity, insulin resistance and fibrinogenesis. What are the systemic effects of this overexpression? Are the muscle and adipose tissues also sensitized to insulin? Or are the observed effects solely due to the liver?

To evaluate peripheral insulin sensitivity, we collected the epididymal white adipose tissue and skeletal muscle of liver-specific Phf2 overexpressing mice fed for 12 weeks on chow or HFHSD. We then conducted western blots analysis in those tissues to characterize the activity of the PI3K/Akt signaling pathway. These data are now included in the Supplementary Figure 3G and Supplementary Figure 9B of the revised

manuscript.

These data demonstrate that there is no change in the PI3K/Akt signaling pathway in WAT and skeletal muscle after liver-specific Phf2 overexpression as compared to GFP mice fed on chow diet (Supplementary Figure 3G). However, liver-specific Phf2 overexpression protects mice from diet-induced obesity and peripheral insulin resistance. Indeed, WAT and skeletal muscle were protected from developing insulin resistance as compared to GFP mice fed on HFHSD as revealed by sustain activation of the PI3K/Akt signaling in those tissues (Supplementary Figure 9B).

3. Phf2 expression also appears to be significant in the pancreas, adipose tissue and muscle – have the authors assessed if a similar regulation of occurs in these tissues either in vitro or in vivo, as this would be informative.

We do not assess the role of Phf2 in other tissues than liver. No data are currently available on the role of Phf2 in muscle or pancreas. However, as mentioned in the introduction and discussion of our manuscript, the function of Phf2 was recently study in adipocytes, in which Phf2 controls adipogenesis and fat storage through the regulation of CEBP α and PPAR γ transcriptional activities {Lee, 2014 #1067}. In addition, mice with targeted disruption of Phf2 display a reduction of their white adipose tissue mass as a result of reduced PPAR γ activity {Okuno, 2013 #955}.

4. Is Phf2 also known to methylate proteins? If so, do the authors think that some of this signaling could also feed into the observed effects? A discussion on this would be useful.

Indeed, as mentioned by the reviewer, there are now several examples where jmjC domain-containing histone demethylases with previously defined roles in histone demethylation also appear to demethylate non-histone proteins to regulate their abundance, stability or activity. This realization that jmjC domain-containing demethylases potentially play widespread roles in protein demethylation raises an important question of whether the primary biological functions is currently attributed to demethylase reactions toward histones or other uncharacterized non-histone proteins. However, to our knowledge, there is no reports today showing that the KDM7 histone demethylase family and more particularly Phf2 can demethylate non-histone proteins.

To address this potential non-histone protein demethylation effect in our phenotype, we now have overexpressed, in primary cultured hepatocytes, either the WT form of Phf2 or its W29A mutant. This W29A mutation, localized within Phf2's PHD domain, has been previously shown to abolish H3K4me3 binding of Phf2 to the promoter of its target genes without affecting its histone demethylase activity {Wen, 2010 #959}. These results are now presented in the supplementary figures 6 and 7 of the revised manuscript.

Overall, ChIP experiments, performed in cultured hepatocytes, confirmed that Phf2 W29A is no longer recruited on the promoter of ChREBP-regulated genes compared with Phf2 WT (supplementary Figure. 6A). At the chromatin level, H3K9me2

demethylation at the SCD1 promoter was not increased by Phf2 W29A overexpression compared to Phf2 WT (supplementary Figure. 6B). Consistently, both chromatin accessibility and ChREBP recruitment at the SCD1 promoter were not increased by Phf2 W29A overexpression (supplementary Figure. 6C and D). As a consequence, Phf2 W29A is unable to enhance the expression of glycolytic and lipogenic genes and increase hepatocyte TG content (supplementary Figure. 6E and F). The fact that Phf2 W29A conserved its histone demethylase activity towards recombinant proteins (supplementary Figure. 6G), demonstrates that Phf2 contributes to the regulation of ChREBP function by erasing H3K9me2 methyl-marks at the promoter of ChREBP target genes.

In the same line of evidence, Phf2 action in regulating Nrf2 transcriptional activity is also directly dependent on its H3K9me2 histone demethylase activity, since Phf2 W29A is unable to enhance Nrf2 activity and protect hepatocytes from palmitate-induced ROS production and hepatocyte apoptosis (supplementary Figure. 7B-G).

Overall, these results clearly demonstrate that H3K9me2 histone demethylase activity is essential for Phf2 beneficial effect in regulating SCD1-mediated SFA desaturation and oxidative stress defense, ruling out non-histone protein demethylation in these processes.

5. In Figure 1E, are there certain histone marks that do not change? If there are several being altered (such as H3K4me3), it is unclear if the transcription of the gene corresponds solely to H3K9me2 demethylation.

To test if certain histone methyl-marks do not change upon Phf2 overexpression, we checked, *in vivo*, levels of H3K9me2, H3K9me1, H3K9me3 and H3K4me3 methylation marks on the promoter of SCD1 and Nrf2 after Phf2 overexpression. As expected, Phf2 overexpression increases H3K9me2 demethylation at the promoter of SCD1 and Nrf2. However, H3K9me1 and H3K9me3 levels were not significantly altered after Phf2 overexpression, supporting a specific action of Phf2 on these two promoters. In addition, H3K4me3 methyl-mark, which signed active promoters, was increased in response to Phf2 overexpression, which correlated with enhanced SCD1 and Nrf2 expression (Supplementary Figure. 3A and B and Figure. 5A).

Despite the fact that certain histone methyl-marks does not change after Phf2 activation, it is however clear in our model that the regulation of glycolytic, lipogenic or anti-oxidative genes are not solely dependent on H3K9me2 histone demethylation but instead the consequence of a complex change in histone methylation profile, which is the result of coactivators and corepressors exchange at the chromatin that ultimately allows the recruitment of the transcriptional machinery. Nevertheless, our study demonstrates that Phf2 H3K9me2 histone demethylase activity is absolutely required to regulate ChREBP and Nrf2 transcriptional activity on the promoter of their target genes as previously mentioned in our response to reviewer#2 (question 4).

However, due to the large amount of data already presented in our study, we decided not include this new set of results in our revised manuscript. We do believe that these results indeed do not bring additional information about the role of Phf2 in the control of gene expression based on its specific H3K9me2 histone demethylase activity (see Figure 1). However, if the Reviewers or the Editor believe that this panel is important, we will include this panel in the revised manuscript.

6. Are the same effects of Phf2 overexpression in vivo in Fig 3 also observed in primary hepatocytes?

In Figure 5 of our revised manuscript, Phf2 was overexpressed in primary cultured hepatocytes. In this set of experiments, which was design to study the role of Phf2 in the regulation of anti-oxidative stress response after palmitate treatment, we were able to recapitulate the effects observed in vivo in Figure 3. Thereby, Phf2 overexpression enhanced, in an Nrf2-dependent manner, the anti-oxidative stress program by increasing GSS and NADPH synthesis in hepatocytes (Figure. 5B and C). More importantly, Phf2 overexpression, by increasing the expression of ROS-scavenger proteins protects hepatocytes from palmitate-induced ROS production and hepatocyte apoptosis. In addition, these effects were dependent on Nrf2, since Nrf2 silencing abolished Phf2-mediated protection from palmitate treatment (Figure. 5B, C and D).

Reviewer #3 (Remarks to the Author):

We would like to thank Reviewer #3 for her/his positive and constructive comments. We have now perform additional experiments and answered the points raised.

In this manuscript, the authors demonstrate the metabolic reprogramming through Plant homeodomain finger 2 (Phf2). They identified Phf2, a histone demethylase for H3K9me2, as a co-activator of carbohydrate-responsive element binding protein (ChREBP). Ectopic expression of Phf2 in mouse livers induces gene expression of proteins encoding free fatty acid metabolism, and it also directs glucose into pentose-phosphate pathway and glutathione synthesis through activation of NF-E2-related factor 2. As a result, mice expressing Phf2 in their livers exhibit steatosis associated with a hyper insulin sensitivity state without inflammation. Interestingly, NAFLD progression by high fat and high sucrose diet was greatly suppressed by Phf2-overexpression, which is

an Nrf2-activation dependent manner. Further, the authors confirmed the increased level of Phf2 in the livers of obese patients with benign hepatic steatosis. Taken together, the authors concluded that Phf2 protects liver from NAFLD-progression. Below I list a few specific points that should help improve the manuscript by making the dataset more consistent with the claims made by the authors.

Specific comments

1. The authors should perform a tracer study with [13C6] glucose in GFP- or Phf2-expressing wild-type and Nrf2-deficient hepatocytes and present the amount of each labelled metabolite.

We agree with the reviewer that performing fluxomic studies using [13C6] glucose in GFP or Phf2 expressing wild type and Nrf2-deficient hepatocytes would bring interesting information about general metabolic reprogramming in a context of Nrf2 deficiency. However, this experiment would take at least 8 months to be completed. In addition, we do not have in our institute the capacity to performed 13C fluxomic analysis and we were unable to find collaborators to set up this experiment within the revision period. Given the large amount of data already present in our revised manuscript, we believe that the requested experiment is beyond the scope of our present study and perhaps best suited in a follow up study.

2. The level of Nrf2 protein is mainly regulated by post-translational modification (i.e., Keap1-mediated ubiquitination of Nrf2 and subsequent degradation by the 26S proteasome). Nevertheless, the authors claim simple up-regulation of Nrf2 through Phf2 leads to its activation. The authors should investigate whether ectopic expression of Phf2 has an effect on the Nrf2-activation at the post-translational level or not.

We agree with the reviewer by the fact that Nrf2 transcriptional activity, independently of the newly identified regulation of its expression by Phf2, is mainly regulated by post-translational modification through ubiquitination and subsequent degradation. Indeed, the Keap1-Nrf2 system is currently recognized as one of the major cellular defense mechanisms against oxidative stress. Under basal conditions, the transcription factor Nrf2 is constitutively degraded through the ubiquitin-proteasome pathway; its binding partner, Keap1, is an adaptor of the ubiquitin ligase complex that targets Nrf2 to the proteasome. However, exposure to reactive oxygen species instigates modification of specific cysteine residues on Keap1, leading to its inactivation. Consequently, Nrf2 is stabilized, and translocates to the nucleus to induce the transcription of numerous anti-oxidative genes.

However, besides this canonical pathway, the autophagy-adaptor p62 interacts with the Nrf2-binding site of Keap1 and competitively inhibits the Keap1-Nrf2 interaction. In this setting, it has recently been described that S351 of p62 is also phosphorylated in a PI3K/Akt/mTORC1-dependent manner, causing p62's affinity for Keap1 to rise. As a result, Nrf2 is stabilized, and it then translocates into the nucleus to induce its cytoprotective targets. In normal cells, this functional interaction serves as a host defense mechanism, leading to expression of antioxidant and anti-inflammation

enzymes and subsequent selective cleanup of cytotoxic structures {Komatsu, 2010 #1064; Lau, 2010 #1065}.

Interestingly, in our model, liver-specific Phf2 overexpression enhanced mTORC1 activity as revealed by increased p70S6K and S6K phosphorylation (Figure. 2G). As a result p62 phosphorylation at S351 is enhanced, which correlates with Nrf2 stabilization and activation (Figure. 5A). Altogether, our results demonstrate that, in addition to its direct effect on Nrf2 expression, Phf2 overexpression, by stimulating the PI3K/Akt/mTORC1 signaling pathway, may thus further participate in the stabilization and induction of Nrf2 and help the expression of genes involved in the pentose phosphate pathway, glutathione biosynthesis and anti-oxidative stress response.

We now discuss this mechanism of Phf2-mediated Nrf2 activation in the Discussion section of our revised manuscript.

3. Does gene silencing or targeting of Phf2 in mouse livers promote NAFLD-progression? This reviewer thinks that the authors can conduct such experiments using adenovirus system or hydrodynamic gene transfer.

In the revised version of our manuscript, we now have included experiments showing that liver-specific Phf2 deficiency, in opposite to Phf2 overexpression, enhanced the entry of NAFLD into fibrosis when mice are fed on HFHSD (Figure. 7).

Indeed, having seen that Phf2 expression was gradually decreased in the liver of mice fed a HFHSD diet when compared to mice fed on a normal chow diet (NCD) (Figure. 7A), we decided to stably inhibit Phf2 expression specifically in the liver of C57Bl6/J mice through the use of an associated adenovirus (AAV) strategy (Figure. 7B-H). We then determine its consequences in term of susceptibility to fibrosis development. To achieve this specificity, an unspecific shRNA (USi) or Phf2 shRNA (Phf2i) was expressed under the control of the albumin promoter. As a result, Phf2 silencing enhanced liver fibrosis when compared to control mice upon HFHSD feeding (Figure. 7B and C). Likewise, expression of Col1a1, TIMP1 and α -SMA were increased in Phf2i mice compared to USi mice along with collagen deposition, number of fibrotic area and hepatocyte apoptosis (Figure. 7C, D and E). In addition, serum levels of ALAT and ASAT were further increased in Phf2i mice (Figure. 7F). Phf2 silencing also increases liver oxidative damages compared to USi mice facilitating the conversion of NAFLD into fibrosis upon HFHSD feeding (Figure. 7G and H).

For this additional panel, based on the question raised by Reviewer#3, we only focus our attention to clarify the contribution of Phf2 deficiency to NAFLD progression. As a consequence, in the present revised manuscript, we decided to not present data showing the consequences of Phf2 silencing on inflammation, glucose tolerance or insulin resistance upon HFHSD feeding. In contrast, we chose to present the protective effects of Phf2 overexpression on inflammation and glycemic control in the context of HFHSD feeding, which we believe bring crucial information about the contribution of Phf2 activation to the physiopathology of obesity and type 2 diabetes.

These results on inflammation, glucose tolerance and insulin sensitivity, that we do not wish to include in the revised version of this manuscript, are summarized below:

Overall, Phf2 silencing enhances the pro-inflammatory response upon HFHSD feeding as revealed by increased expression of IL6, IL1 β and TNF α .

*P < 0.01 USi HFHSD compared to USi NCD, **P < 0.01 USi HFHSD compared to Phf2i HFHSD.

In the same line of evidence, Phf2 silencing further enhances the glucose intolerance and insulin resistance upon HFHSD feeding as compared to USi mice on the same diet.

*P < 0.01 USi HFHSD compared to Phf2i HFHSD, **P < 0.05 USi HFHSD compared to Phf2i HFHSD.

4. The molecular mechanism by which Phf2 activates PI3K/Akt pathway remains unclear.

As lipotoxicity and insulin resistance is generally attributed to SFA, their conversion into specific MUFA, in response to Phf2 overexpression, could be directly instrumental in the protective effect of Phf2 from insulin resistance, by reducing the intracellular concentration of SFA. To demonstrate the consequences of Phf2-mediated SFA desaturation in these effects, we now have included in our revised manuscript, a

lipidomic analysis performed on primary cultured hepatocytes overexpressing Phf2 and in which SCD1 expression was inhibited (Figure. 2J). We also characterized more precisely the pro-inflammatory signaling pathway by western blots (Figure. 2I).

In control hepatocytes, palmitate treatment increases the intracellular content of SFA, which is correlated with the induction of pro-inflammatory signaling pathway and development of insulin resistance, as evidence by increased JNK, IKKb or NFkB phosphorylation and decreased AKT or ERK phosphorylation (Figure. 2I and K). However, Phf2 overexpression, by increasing SFA desaturation, decreases SFA content in hepatocytes after palmitate treatment. As a consequence, Phf2 overexpression prevents palmitate-induced inflammation and insulin resistance. In contrast, in a context of SCD1 deficiency, Phf2 overexpression is no longer able to decrease SFA content in hepatocytes by converting them into MUFA. Consequently, SFA content is significantly increased under palmitate treatment (Figure. 2J). In this context, inflammation and insulin resistance are enhanced even in the setting of Phf2 overexpression (Figure. 2I and K).

Overall, our results clearly demonstrate that this is not the MUFAs per se that would prevent from inflammation and insulin resistant but likely the decrease in SFA content after their specific desaturation by SCD1.

5. In human obese patients, the level of Phf2 protein fluctuated on the disease-state. Do the authors have any idea to address this?

We agree with the reviewer that Phf2 protein levels indeed fluctuated during NAFLD progression in humans. However, the regulation of Phf2 expression and/or Phf2 protein stability is absolutely not known to date. In general, the regulation of jmjC domain-containing demethylase expression during the pathogenesis of obesity, type 2 diabetes and even cancer as not been studied to our knowledge.

Overall, despite the fact that this question of the regulation of Phf2 expression and/or activity is important, we believe that the requested experiment is beyond the scope of our present study. Indeed, the main objective of our study aimed to provide the proof of concept that deregulation of Phf2 activity was able to promote the conversion of simple hepatic steatosis to NASH and fibrosis and not necessarily to document how Phf2 expression or activity is regulated during the disease progression. This important question will be perhaps best suited in a follow up study.

Minor points

We would like to thank the reviewer for pointing all of these mistakes in the manuscript. We have appropriately corrected these inaccuracies in the text.

1. Page 7, 11 line from the bottom. Replace “Figure 2D” by “Figure 2C”.
2. Page 8, line 13. “Figure 3I” should be “Figure 2I”.
3. Figure legend. Figure 1, line 5. Typo: substitute “transected” with “transfected”
4. Figure legend. Figure 6, title. Substitute “fribrogenesis” with “fibrogenesis”.

5. Bar graphs shown in Fig. 4C, E and Fig. 5G are difficult to identify. Change the color of bar.
6. In Fig. 5G, is a right graph showing ARE-luc activity correct?

We would like to sincerely thank the reviewer for pointing this important mistake in the graph showing ARE-luc activity in response to glucose stimulation, which was performed in primary cultured hepatocytes. In the initial version of our manuscript this graph was indeed a duplication of the panel presented in Figure 5I. We have appropriately corrected this panel in the revised version of our manuscript.

Reviewers' comments:

Reviewer #1 (Remarks to the Author):

Major points

1. It is unclear under which conditions Akt/PI3K phosphorylation was measured. Still there is no gold standard proof of effects or absence of effect on insulin sensitivity in the different tissues. It would have been useful to perform clamp studies.

2. I did not find the method for separate analysis of DAG in different compartments. Furthermore, the absence of PKC activity with the assay employed is not convincing, did the authors perform a positive control test, showing that this assay identifies physiological PKC epsilon activation? Finally, I do not understand the statement on proinflammatory response in this context. If the authors assume that SFA play the key role, did they assess ceramides?

Reviewer #2 (Remarks to the Author):

The authors have addressed the previous critiques with new experimental data or explanations in the text that support the conclusions.

Reviewer #3 (Remarks to the Author):

The authors addressed all concerns raised by my previous review.

A minor comment.

Originally, Ichimura et al reported phosphorylation of p62 at S351 followed by the activation of Nrf2 (Mol. Cell 2013). Cite this paper.

Reviewers' comments:

Reviewer 1

Major points

1. It is unclear under which conditions Akt/PI3K phosphorylation was measured. Still there is no gold standard proof of effects or absence of effect on insulin sensitivity in the different tissues. It would have been useful to perform clamp studies.

We agree with Reviewer 1 on the fact that our experimental conditions were not very well described in the method section and figure legend. We now have addressed this concern by describing in more details the conditions under which the PI3K/Akt signaling was measured (legend of Supplementary Figure 3 (panel G)).

For insulin signaling experiments in skeletal muscles and white adipose tissue (Supplementary Figure 2G), mice overexpressing either GFP or Phf2 were fasted overnight and then were injected with PBS or 1 unit of regular human insulin/kg (Actrapid Penfill, NovoNordisk) via the portal vein. Three minutes after the insulin bolus, skeletal muscle and white adipose tissue were removed and snap frozen in liquid nitrogen. In the previous version of our manuscript, the PI3K/Akt signaling was then analyzed by western blot on insulin-stimulated conditions only (Supplementary Figure 2G).

In this new version of our manuscript, we have substituted our previous panel 2G showing the PI3K/Akt signaling in insulin-stimulated WAT and skeletal muscle with western blots analysis using both PBS and insulin-stimulated tissues. This new panel (Supplementary Figure 2G) clearly demonstrates that insulin stimulation significantly increased Akt, p70S6K and GSK3 β phosphorylation as compared to PBS treatment in both fasted GFP or Phf2 mice. Again, as shown in Supplementary Figure 2G, there is no apparent difference in the PI3K/Akt signaling in skeletal muscle and white adipose tissue after insulin treatment between liver-specific GFP or Phf2 overexpressing mice, indicating an absence of change in peripheral insulin sensitivity. Moreover, supporting this observation, there is no change in skeletal muscle glycogen content in fed Phf2 mice compared to GFP (Supplementary Table 1).

Based on these data, we are not convincing that clamp studies will bring a different conclusion regarding peripheral insulin sensitivity. In addition, this question is not the main objective of our study, which was to determine at the molecular level the contribution of the histone demethylase Phf2 to the development and progression of NAFLD, in order to connect epigenetic modifications to the physiopathology of obesity. We have discussed the lack of clamp studies in the discussion of our manuscript accordingly, as requested by the editorial board of Nature Communications.

2. I did not find the method for separate analysis of DAG in different compartments. Furthermore, the absence of PKC activity with the assay employed is not convincing, did

the authors perform a positive control test, showing that this assay identifies physiological PKC epsilon activation? Finally, I do not understand the statement on proinflammatory response in this context. If the authors assume that SFA play the key role, did they assess ceramides?

We apologize since the method for separating DAG from different compartments was indeed missing in our revised manuscript. You will find below this method that has been included in the supplementary method of our manuscript.

Hepatic DAG content was separated into membrane and cytoplasmic fractions as previously described (1). Briefly, liver samples (250mg) were homogenized on ice using a dounce homogenizer in lysis buffer containing 10 mM Tris base, 0.5 mM EDTA, 250 mM sucrose, and protease inhibitors. Then 3% sucrose was layered on top of the homogenate, and samples were centrifuged at 100,000g for 1h at 4 °C. The supernatant and lipid layers were removed and designated as the cytoplasmic fraction. The pellet, designated as the membrane fraction, was resuspended in homogenization buffer for DAG analysis. Concentrations of DAG in each fraction were determined by using mass spectrometry analysis as described in our manuscript.

Concerning PKC epsilon activity, we indeed used one positive control to validate PKC epsilon activity measurement. We used protein lysates from liver of C57Bl6/J mice fed for 18 weeks with either a standard diet (NCD) or with a high fat and high sucrose diet (HF/HS) as positive control. This positive control was not included in the previous version of our manuscript. As shown is Figure 2B of our revised manuscript this HFHS diet significantly induces PKC epsilon activity as compared to NCD fed mice. This control, which validate our assay, has been now included in Figure 2 (panel B) of our revised manuscript.

Concerning the last point raised by reviewer 1, we did not see any changes in total liver ceramide content between GFP or Phf2 overexpressing mice (Supplementary Table 1). Ceramide profiling was performed by metabolon, Inc. (Durham, North Carolina, USA) from 30 mg of frozen liver tissue. These data are now included in our revised manuscript (Supplementary Table 1). Even if ceramides are thought to be among the most pathogenic of all lipid species because they promote inflammatory signaling cascades and impair insulin signaling, the fact that ceramides content did not significantly change after Phf2 overexpression did not prompt us to further study their contribution to Phf2 beneficial effect.

Our lipidomic analysis on ceramide species further revealed that ceramides C16:0 were even decreased in Phf2 mice compared to GFP. In contrast, ceramides C18:1 and C24:1 were significantly increased further reinforcing the concept that Phf2-mediated SCD1 regulation and subsequent SFA desaturation in DAG, TG cholesterol esters or ceramides is instrumental in the beneficial outcome on inflammation and insulin resistance. Lipidomic analysis on ceramides has been included in supplementary Figure 3H of our revised manuscript. Overall, these data on ceramides do not change our

conclusion regarding the importance of Phf2-mediated SFA desaturation in its protective effect against inflammation and insulin resistance.

References.

1. Cantley JL, Yoshimura T, Camporez JP, Zhang D, Jornayvaz FR, Kumashiro N, Guebre-Egziabher F, Jurczak MJ, Kahn M, Guigni BA, et al. CGI-58 knockdown sequesters diacylglycerols in lipid droplets/ER-preventing diacylglycerol-mediated hepatic insulin resistance. *Proc Natl Acad Sci U S A*. 2013;110(5):1869-74.

Reviewer 3

The authors addressed all concerns raised by my previous review.

A minor comment.

Originally, Ichimura et al reported phosphorylation of p62 at S351 followed by the activation of Nrf2 (Mol. Cell 2013). Cite this paper.

We have cited the study of Ichimura et al accordingly to acknowledge the role of p62 S351 phosphorylation in the control of nrf2 activation.

REVIEWERS' COMMENTS:

Reviewer #1 (Remarks to the Author):

My comments have been adequately addressed and I have no further comments